



# Nitrate radical generation via continuous generation of dinitrogen pentoxide in a laminar flow reactor coupled to an oxidation flow reactor

Andrew T. Lambe[1], Ezra C. Wood[2], Jordan E. Krechmer[1], Francesca Majluf[1], Leah R. Williams[1], Philip L. Croteau[1], Manuela Cirtog[3], Anaïs Féron[3], Jean-Eudes Petit[4], Alexandre Albinet[5], Jose L. Jimenez[6], and Zhe Peng[6]

[1]Center for Aerosol and Cloud Chemistry, Aerodyne Research Inc., Billerica, MA, USA
[2]Dept. of Chemistry, Drexel University, Philadelphia, PA, USA
[3]Laboratoire Inter-Universitaire des Systèmes Atmosphériques (LISA), UMR CNRS 7583, Université Paris-Est-Créteil, Université de Paris, Institut Pierre Simon Laplace (IPSL), Créteil, France
[4]Laboratoire des Sciences du Climat et de l'Environnement (CNRS-CEA-UVSQ), CEA Orme des Merisiers, Gif-sur-Yvette, France
[5]Institut National de l'Environnement Industriel et des Risques (Ineris), Verneuil-en-Halatte, France
[6]Dept. of Chemistry and Cooperative Institute for Research in Environmental Sciences (CIRES), University of Colorado, Boulder, CO, USA

**Correspondence:** Andrew T. Lambe (lambe@aerodyne.com), Zhe Peng (zhe.peng@colorado.edu)

**Abstract.** Oxidation flow reactors (OFRs) are an emerging tool for studying the formation and oxidative aging of organic aerosols and other applications. The majority of OFR studies to date involved generation of the hydroxyl radical (OH) to mimic daytime oxidative aging processes. On the other hand, use of the nitrate radical ($NO_3$) in modern OFRs to mimic nighttime oxidative aging processes has been limited due to the complexity of conventional techniques that are used to generate $NO_3$.

Here, we present a new method that uses a laminar flow reactor (LFR) to continuously generate dinitrogen pentoxide ($N_2O_5$) in the gas phase at room temperature from the $NO_2 + O_3$ and $NO_2 + NO_3$ reactions. The $N_2O_5$ is then injected into a dark Potential Aerosol Mass OFR and decomposes to generate $NO_3$; hereafter, this method is referred to as "OFR-i$N_2O_5$" ("i" = injected). To assess the applicability of the OFR-i$N_2O_5$ method towards different chemical systems, we present experimental and model characterization of the integrated $NO_3$ exposure, $NO_3$:$O_3$, $NO_2$:$NO_3$, and $NO_2$:$O_2$ as a function of LFR and OFR

conditions. These parameters were used to investigate the fate of representative organic peroxy radicals ($RO_2$) and aromatic alkyl radicals generated from volatile organic compound (VOC) + $NO_3$ reactions, and VOCs that are reactive towards both $O_3$ and $NO_3$. Finally, we demonstrate the OFR-i$N_2O_5$ method by generating and characterizing secondary organic aerosol from the $\beta$-pinene + $NO_3$ reaction.

## 1 Introduction

The importance of nitrate radicals ($NO_3$) as a nighttime oxidant is well established (Wayne et al., 1991; Brown and Stutz, 2012; Ng et al., 2017). In the atmosphere, $NO_2 + O_3$ is the primary source of $NO_3$, after which $NO_3$ exists in equilibrium with





$NO_2$ and $N_2O_5$. Atmospheric nighttime $NO_3$ mixing ratios can vary by at least two orders of magnitude, ranging from 1 ppt or less in remote areas to 10-400 ppt in polluted urban regions (Finlayson-Pitts and Pitts Jr., 2000; Asaf et al., 2010; Warneck and Williams, 2012; Ng et al., 2017). Atmospheric organic compounds that are reactive towards $NO_3$ include isoprene and monoterpenes that are emitted from biogenic sources (including urban vegetation); phenols and methoxyphenols emitted from

biomass burning; and polycyclic aromatic hydrocarbons (PAHs) emitted from combustion processes. $NO_3$ oxidation of these compounds generates oxygenated volatile organic compounds (OVOCs) and/or secondary organic aerosol (SOA), including particulate organic nitrates or nitroaromatics. The importance of these sources and processes are likely to continue to increase for the foreseeable future due to climate change (Melaas et al., 2016; Short, 2017).

Laboratory studies have attempted to elucidate the mechanisms associated with $NO_3$-initiated oxidative aging processes, in
the gas and condensed phase, and in environmental chambers and flow tubes. Traditional $NO_3$ generation techniques typically utilize $N_2O_5$ as the radical precursor. $N_2O_5$ is generated from the reaction $NO + O_3 \rightarrow NO_2 + O_2$, followed by the reactions $NO_2 + O_3 \rightarrow NO_3 + O_2$ and $NO_2 + NO_3 \rightarrow N_2O_5$. The synthesized $N_2O_5$ is collected and stored in a cold trap under dry conditions to minimize hydrolysis of $N_2O_5$ to nitric acid ($HNO_3$). This method has limitations that hinder widespread usage: specifically, long-term storage and handling of $N_2O_5$ at low temperature and dry conditions is difficult, and continuous
generation of $N_2O_5$ as is required for oxidation flow reactors (OFRs) or other continuous flow chambers is challenging. Field studies investigating the $NO_3$-induced SOA formation potential of ambient air are thus extremely limited (Palm et al., 2017). Alternative $NO_3$ generation techniques that utilize reactions between chlorine atoms and chlorine nitrate ($ClONO_2$) or fluorine atoms and $HNO_3$ require cold storage of $ClONO_2$ and handling or generation of reactive halogen species that are reactive towards organic compounds (Burrows et al., 1985).

To address issues associated with traditional $NO_3$ generation techniques, we developed and characterized a new method that is well suited to applications where a continuous source of $N_2O_5/NO_3$ is required, such as OFR studies. The method is capable of continuous $N_2O_5$ generation in the gas phase at room temperature using a laminar flow reactor (LFR) that is coupled to a dark OFR in which $N_2O_5$ injected into the OFR decomposes to generate $NO_3$ and initiate oxidation of reactive VOCs. Hereafter, we refer to this method as "OFR-i$N_2O_5$" ("i" = injected"). We present experimental and model characterization
of OFR-i$N_2O_5$ as a function of LFR and OFR conditions, and we demonstrate application of OFR-i$N_2O_5$ to generate and characterize SOA from the $\beta$-pinene + $NO_3$ reaction.

## 2   Methods

### 2.1   $N_2O_5$ and $NO_3$ generation

Figure 1 shows a process flow diagram of the OFR-i$N_2O_5$ method. Separate flows containing $NO_2$ and $O_3$ were input to a PFA
tube with 2.54 cm outer diameter, 2.22 cm inner diameter, and 152.4 cm length that was operated as a LFR (Wood et al., 2003; Boyd et al., 2015). A compressed gas cylinder containing $1.00 \pm 0.02\%$ $NO_2$ in $N_2$ (Praxair) was used to supply $NO_2$. While not used for this study, replacing $NO_2$ with NO to avoid $NO_2$-to-$HNO_3$ conversion inside the gas cylinder and increasing $[O_3]$ accordingly achieves similar results. $O_3$ was generated by passing 1750-1800 $cm^3$ $min^{-1}$ of pure $O_2$ through a custom





$O_3$ chamber housing a mercury fluorescent lamp (GPH212T5VH, Light Sources, Inc.) or 500-1800 cm$^3$ min$^{-1}$ $O_2$ through a corona discharge ozone generator (Enaly 1KNT). We used 1800 cm$^3$ min$^{-1}$ of $O_2$ carrier gas flow through the LFR (Re $\sim$ 110, i.e. laminar flow) to achieve $\tau_{LFR}$ = 20 s for reasons that are discussed in Section 3.1. The $NO_2$ mixing ratio entering the LFR, $[NO_2]_{0,LFR}$, was calculated from the $NO_2$ mixing ratio in the compressed gas mixture and the dilution ratio of 0-50

5    cm$^3$ min$^{-1}$ or 0-1300 cm$^3$ min$^{-1}$ gas flow into $O_2$ which was controlled using mass flow controllers. The $O_3$ mixing ratio entering the LFR, $[O_3]_{0,LFR}$, was measured using a 2B Technologies 106-MFT or a Teledyne M452 flow-through $O_3$ analyzer when generated from the mercury lamp or corona discharge source respectively. The output of the LFR was mixed with a carrier gas containing 3.8 L min$^{-1}$ synthetic air and then injected into a Potential Aerosol Mass OFR (Aerodyne Research, Inc.), which is a horizontal 13.3 L aluminum cylindrical chamber operated in continuous flow mode (Kang et al., 2007; Lambe

et al., 2011, 2019) with 6.5 L min$^{-1}$ flow through the reactor. The mean residence time in the OFR ($\tau_{OFR}$) was 120 $\pm$ 34 s ($\pm$ 1$\sigma$), as obtained from measurements of 10 s pulsed inputs of $NO_2$ to the OFR obtained using a 2B Technologies Model 405 $NO_x$ analyzer (Figure S1). Across all experiments, the relative humidity in the OFR (RH$_{OFR}$) was controlled in the range of 7-85% at 23-25° C by passing the carrier gas through a Nafion humidifier (Perma Pure LLC) or heated recirculating water bath (Neslab Instruments, Inc.) prior to mixing with the LFR outflow. The $O_3$ mixing ratio at the exit of the OFR was measured

with a 2B Technologies Model 106-M ozone analyzer.

### 2.1.1   OFR-iN$_2$O$_5$ characterization studies

In one set of experiments, the integrated $NO_3$ exposure ($NO_{3exp}$), defined here as the product of the average $NO_3$ concentration and $\tau_{OFR}$, was characterized by measuring the decay of VOC tracers reactive towards $NO_3$ using a Tofwerk/Aerodyne Vocus Proton Transfer Reaction Time-of-Flight Mass Spectrometer (PTR-MS) (Krechmer et al., 2018). For this purpose, the

tracer decay method is advantageous to direct $NO_3$ measurements at the OFR inlet and/or outlet because potential $NO_3$ concentration gradients inside the OFR that might otherwise bias $NO_{3exp}$ are easily accounted for. Tracers that were liquid at room temperature were injected into the OFR through a 10.2 cm length of 0.0152 cm ID Teflon tubing at a liquid flow rate of about 0.94 $\mu$L hr$^{-1}$ using a syringe pump, prior to evaporation into a 2.4 L min$^{-1}$ $N_2$ carrier gas. In preliminary studies, tracers such as isoprene and $\beta$-pinene were too reactive towards $NO_3$ to facilitate accurate characterization of $NO_{3exp}$ over the majority

of OFR-iN$_2$O$_5$ conditions that were investigated. Thus, experiments described in this paper used mixtures of tracers with bimolecular $k_{NO_3}$ ranging from approximately $10^{-16}$ to $10^{-13}$ cm$^3$ molecules$^{-1}$ s$^{-1}$ and $k_{O_3}$ < $10^{-19}$ cm$^3$ molecules$^{-1}$ s$^{-1}$ (Table S1). Acetonitrile was used as a nonreactive tracer. In "low $O_3$" experiments ($[O_3]_{0,LFR}$ = 10 to 300 ppm) a mixture of acetonitrile, butanal, thiophene, 2,3-dihydrobenzofuran, and naphthalene-d$_8$ (C$_{10}$D$_8$), each with mixing ratios of approximately 660 ppb, 50 ppb, 56 ppb, 40 ppb, and 18 ppb, respectively, was used. For this tracer mixture, the total external $NO_3$

reactivity ($NO_3R_{ext}$), which is the summed product of each tracer mixing ratio and its $NO_3$ rate constant, was approximately 0.07 s$^{-1}$. Naphthalene-d$_8$ was introduced by flowing 5 cm$^3$ min$^{-1}$ $N_2$ through a Teflon tube packed with solid C$_{10}$D$_8$. In "high $O_3$" experiments ($[O_3]_{0,LFR}$ = 6100 to 7400 ppm), which generated higher $NO_{3exp}$, a mixture of acetonitrile (275 ppb), toluene (45 ppb), o-xylene (40 ppb), p-cymene (31 ppb), 1,2,4-trimethylbenzene (35 ppb), 1-butanol (53 ppb), benzaldehyde (47 ppb), butanal (53 ppb), and thiophene (56 ppb) was used, with $NO_3R_{ext} \approx 0.38$ s$^{-1}$.





In another set of experiments that were conducted as part of the Aerosol Chemical Monitor Calibration Center (ACMCC) particulate organonitrates (pON) experiment (Albinet et al., 2019), direct measurements of $NO_3$ generated via OFR-i$N_2O_5$ were performed using a newly developed Incoherent Broad Band Cavity Enhanced Absorption Spectroscopy (IBBCEAS) technique (Cirtog et al., manuscript in preparation). The IBBCEAS instrument that was used measured absorption as a function

of wavelength between $\lambda$ = 640 and 680 nm, thereby allowing simultaneous measurements of $NO_2$ and and $O_3$ along with $NO_3$. During this experiment, pON were generated in a PAM OFR that used $[O_3]_{0,LFR}$ = 150-180 ppm and $[NO_2]_{0,LFR}:[O_3]_{0,LFR}$ = 0.75, 1.0, and 2.0. IBBCEAS has been used to measure trace $NO_3$ levels in laboratory and field studies (Venables et al., 2006; Kennedy et al., 2011) using measurement principles that are described in detail by Romanini et al. (1997) and Langridge et al. (2008). Briefly, measurements were conducted by exciting a high-finesse optical cavity formed by two high reflectivity

mirrors with an incoherent broad-band-source centered on the $\lambda$ = 662 nm absorption cross section of $NO_3$. Photons resonate between the two mirrors, allowing an effective path length of up to 4.5 km inside the cavity. The absorption coefficient of the sample in the cavity, $\alpha(\lambda)$, was calculated using Equation 1:

$$\alpha(\lambda) = \left( \frac{I_0(\lambda)}{I(\lambda)} - 1 \right) \left( \frac{1 - R(\lambda)}{d} \right) \tag{1}$$

Where $I(\lambda)$ and $I_0(\lambda)$ were the measured transmitted intensities in the presence and absence of $NO_3$, respectively, $d$ = 61 cm

was the distance between the cavity mirrors, and $R(\lambda)$ was the mirror reflectivity ($\sim$99.98%), which was measured before each experiment using a certified calibration cylinder containing 600 ppb $NO_2$ in zero air (Air Liquide). $NO_3$ concentrations were calculated from the Beer-Lambert law using the measured $\alpha_{662}$ values and a $NO_3$ absorption cross section of $2 \times 10^{-17}$ cm$^2$ at $\lambda$ = 662 nm (Vandaele et al., 1998; Orphal et al., 2003). To avoid saturation of the IBBCEAS in these experiments, the OFR sample was diluted by a controlled dilution factor ranging from 9 to 50 and the detection response was deliberately lowered by

reducing the optical path length.

To demonstrate the application of OFR-i$N_2O_5$ to generate SOA, the chemical composition and mass concentration of $\beta$-pinene + $NO_3$ condensed-phase oxidation products was measured with an Aerodyne long-time-of-flight aerosol mass spectrometer (L-ToF-AMS) and/or an aerosol chemical speciation monitor (ACSM). A syringe pump was used to deliver $\beta$-pinene (10% (v/v) in carbon tetrachloride or 50% (v/v) in ethanol) into the carrier gas flow at liquid flow rates ranging from 0.94 to 19

$\mu$L hr$^{-1}$. Results presented in this paper assume an AMS/ACSM collection efficiency of 0.5 (Middlebrook et al., 2012) and a relative ionization efficiency of particulate organics equal to 1.6 (Xu et al., 2018).

## 2.2 Photochemical model

We used the KinSim chemical kinetic solver to calculate concentrations of radical/oxidant species (Peng et al., 2015; Peng and Jimenez, 2017, 2019). The KinSim mechanism shown in Table S2 was adapted from Palm et al. (2017) to model $NO_3$ and

30 $N_2O_5$ concentrations in the LFR and OFR. Inputs to the LFR-KinSim model were: $[O_3]_{0,LFR}$, $[NO_2]_{0,LFR}$, RH = 1%, $T$ = 24°C, $\tau_{LFR}$ = 20 s (modeled as plug flow, see Section 3.1), and first-order wall loss rates of $NO_3$ and $N_2O_5$ ($k_{w_{LFR},NO_3}$ and $k_{w_{LFR},N_2O_5}$). Inputs to the OFR-KinSim model were: $[O_3]$, $[NO_2]$, $[NO_3]$, and $[N_2O_5]$ output from the LFR and scaled by a





measured dilution factor of 4.4; RH and $T$ measured in the OFR; $\tau_{\mathrm{OFR}}$ = 120 s, $k_{\mathrm{w_{OFR},NO_3}}$, and $k_{\mathrm{w_{OFR},N_2O_5}}$, and input VOC tracer concentrations and their $k_{\mathrm{NO_3}}$ values. Because the calculated $N_2O_5$ residence time in the OFR inlet ($\sim$0.04 s) was short compared to the $N_2O_5$ decomposition timescale at $T$ = 23 - 25 °C ($\sim$20 s), potential thermal decomposition of $N_2O_5$ during the dilution step was not considered in the model.

5    **2.2.1   LFR and OFR $k_{\mathrm{w,NO_3}}$ and $k_{\mathrm{w,N_2O_5}}$ values**

Published $k_{\mathrm{w,NO_3}}$ values onto Teflon/Pyrex tubing with 1 cm and 4 cm ID are 0.2 and 0.1 $\mathrm{s}^{-1}$ respectively (Dubé et al., 2006; Wood et al., 2003), which bound the 2.22 cm ID of the LFR used in this study. Assuming $k_{\mathrm{w}}$ is inversely proportional to the internal diameter of the tube, we assumed $k_{\mathrm{w_{LFR},NO_3}}$ = 0.15 $\mathrm{s}^{-1}$. Extrapolating this value to the OFR yielded $k_{\mathrm{w_{OFR},NO_3}}$ = 0.02 $\mathrm{s}^{-1}$. At fixed OFR-i$N_2O_5$ conditions, varying $k_{\mathrm{w_{LFR},NO_3}}$ between 0 and 0.2 $\mathrm{s}^{-1}$ changed $\mathrm{NO_{3exp}}$ achieved in the OFR by 10   0.3%. Results were even less sensitive to $k_{\mathrm{w_{OFR},NO_3}}$ assumed for the OFR because of its larger diameter and higher $\mathrm{NO_3R_{ext}}$.

    Published $k_{\mathrm{w,N_2O_5}}$ values onto dry (RH $\approx$ 20%) Pyrex/PFA tubing with 4 and 7 cm ID are 0.04 and 0.009 $\mathrm{s}^{-1}$ respectively (Wagner et al., 2008; Gržinić et al., 2015). Extrapolating these values to the LFR used here yielded $k_{\mathrm{w,N_2O_5}}$ values of 0.07 and 0.03 $\mathrm{s}^{-1}$ respectively. We therefore assumed $k_{\mathrm{w,N_2O_5}}$ = 0.05 $\mathrm{s}^{-1}$ in the LFR-KinSim model. In preliminary OFR-KinSim modeling studies, we assumed $k_{\mathrm{w,N_2O_5}}$ = 0.014 $\mathrm{s}^{-1}$ (Palm et al., 2017). However, as will be discussed in Section 3.3, $k_{\mathrm{w,N_2O_5}}$ 15   was humidity-dependent and required modifications to match measured $\mathrm{NO_{3exp}}$ values as a function of $\mathrm{RH_{OFR}}$.

## 3   Results and Discussion

### 3.1   LFR design considerations

The optimal LFR residence time ($\tau_{\mathrm{LFR}}$) was identified using model simulations of the injection of 300 ppm $O_3$ and $NO_2$ into the LFR followed by dilution and injection of the LFR output into an OFR operated with $\tau_{\mathrm{OFR}}$ = 120 s. Figure S2 plots 20   $\mathrm{NO_{3exp}}$ achieved in the OFR as a function of $\tau_{\mathrm{LFR}}$ ranging from 1 to 60 s. Potential entry length effects that may have influenced results obtained below $\tau_{\mathrm{LFR}} \approx$ 4-5 s were not considered in the model. Figure S2 shows that maximum $\mathrm{NO_{3exp}}$ in the OFR was obtained at $\tau_{\mathrm{LFR}}$ = 20 s at room temperature (unheated case); other $\mathrm{NO_{3exp}}$ values were normalized to this condition. Below $\tau_{\mathrm{LFR}}$ = 20 s, $\mathrm{NO_{3exp}}$ was suppressed due to higher $NO_2$ levels entering the OFR. Above $\tau_{\mathrm{LFR}}$ = 20 s, $\mathrm{NO_{3exp}}$ was suppressed due to lower $N_2O_5$ levels entering the OFR because of more extensive LFR wall loss.

25     In traditional studies of $NO_3$ oxidative aging processes that are conducted at low pressure and short residence time ($\tau \sim$ 1 s), $N_2O_5$ is heated to generate a burst of $NO_3$ prior to injection into the system (Knopf et al., 2011). While not experimentally considered in this work, we modeled the $\mathrm{NO_{3exp}}$ achieved assuming complete thermal dissociation of $N_2O_5$ between the LFR and OFR - for example, by heating to 120°C for 300 ms (Wood et al., 2003). Figure S2 suggests that the effect of heating $N_2O_5$ on $\mathrm{NO_{3exp}}$ was most significant at short $\tau_{\mathrm{LFR}}$, where [$N_2O_5$] at the exit of the LFR was higher due to less wall loss 30   and room-temperature decomposition. For example, at $\tau_{\mathrm{LFR}}$ = 8 s, the modeled $\mathrm{NO_{3exp}}$ was 2.8 times higher in the complete-dissociation case than in the unheated case, whereas $\mathrm{NO_{3exp}}$ increased by factors of 2.3 and 1.5 at $\tau_{\mathrm{LFR}}$ = 20 and 60 s. Thus,





a combination of reducing $\tau_{\mathrm{LFR}}$ and heating $N_2O_5$ at the exit of the LFR increases $NO_{3\mathrm{exp}}$, and should be explored for future advanced implementations of OFR-i$N_2O_5$.

## 3.2    Example OFR-i$N_2O_5$ characterization studies

Figure 2a shows time series of $O_3$ and $NO_2$ concentrations during an OFR-i$N_2O_5$ characterization experiment where $RH_{\mathrm{OFR}}$
= 11%, $[O_3]_{0,\mathrm{LFR}}$ = 280 ppm, and $[NO_2]_{0,\mathrm{LFR}}$ = 0 to 320 ppm. Figure 2b shows time series of acetonitrile ($C_2H_3N$), butanal ($C_4H_8O$), thiophene ($C_4H_4S$), 2,3-dihydrobenzofuran ($C_8H_8O$) and naphthalene-d$_8$ ($C_{10}D_8$) signals measured during the same period. Following $NO_3$ generation, the fractional decay of $C_2H_3N$, $C_4H_8O$, $C_4H_4S$ and $C_8H_8O$ increased with increasing tracer $k_{NO_3}$ as expected. $C_8H_8O$ was too reactive to measure any significant changes in its decay as a function of OFR-i$N_2O_5$ conditions shown in Figure 2; however, maximum decay of $C_4H_8O$ and $C_4H_4S$ was observed at $[NO_2]_{0,\mathrm{LFR}}$:$[O_3]_{0,\mathrm{LFR}}$
$\approx 0.7$ in this experiment. Decay of naphthalene-d$_8$, which was influenced by both $NO_3$ and $NO_2$ concentrations (Table S1), was maximized at $[NO_2]_{0,\mathrm{LFR}}$:$[O_3]_{0,\mathrm{LFR}} \approx 0.3$ to 1.1.

To confirm that VOC degradation shown in Fig. 2b was due to reaction with $NO_3$, Figure 3 shows IBBCEAS measurements of $NO_3$ obtained in separate OFR-i$N_2O_5$ characterization experiments that used $[O_3]_{0,\mathrm{LFR}}$ = 150-160 ppm and $[NO_2]_{0,\mathrm{LFR}}$:$[O_3]_{0,\mathrm{LFR}}$ = 0.75 and 2.0. The maximum IBBCEAS signal observed at $\lambda$ = 662 nm indicated the presence of $NO_3$, as is evident from comparison with the wavelength-dependent absorption cross section of $NO_3$ obtained by Orphal et al. (2003) and plotted in Figure 3b. Additionally, Figure S3 shows the relative rate coefficient obtained from the decay of $C_4H_8O$ and $C_4H_4S$ measured with PTR-MS. We measured a relative rate coefficient of 2.83, which is in agreement with a relative rate coefficent value of $3.22 \pm 0.95$ calculated from $C_4H_8O + NO_3$ and $C_4H_4S + NO_3$ rate coefficients (Atkinson, 1991; D'Anna et al., 2001). Ions corresponding to peroxy butyl nitrate, nitrothiophene, and nitronaphthalene-d$_7$, which are known $NO_3$ oxidation products of $C_4H_8O$, $C_4H_4S$, and $C_{10}D_8$, respectively (Atkinson et al., 1990; Jenkin et al., 2003; Saunders et al., 2003; Cabañas et al., 2005), were also detected with PTR-MS. Tracer decay experiments similar to the measurements shown in Figure 2 were repeated over $[O_3]_{0,\mathrm{LFR}}$ ranging from 10 to 7400 ppm, $[NO_2]_{0,\mathrm{LFR}}$ ranging from 0 to 7200 ppm, and $RH_{\mathrm{OFR}}$ ranging from 7 to 85%. For experiments where $[O_3]_{0,\mathrm{LFR}} > 6000$ ppm, $NO_{3\mathrm{exp}}$ was calculated from the decay of o-xylene because (1) p-cymene has a large ionized fragment at $C_7H_9^+$ (thus interfering with detection of toluene that was also present), (2) $NO_3$ oxidation products were generated that interfered with detection of oxygenated tracers (butanol, benzaldehyde, butanal) and(3) the remaining tracers that were used were too reactive towards $NO_3$ to accurately constrain $NO_{3\mathrm{exp}}$.

## 3.3    Effect of $RH_{\mathbf{OFR}}$, $[O_3]_{0,\mathbf{LFR}}$, and $[NO_2]_{0,\mathbf{LFR}}$ on $NO_{3\mathbf{exp}}$

Figure 4 shows $NO_{3\mathrm{exp}}$ as a function of $RH_{\mathrm{OFR}}$ at $[O_3]_{0,\mathrm{LFR}}$ = 250 ppm and $[NO_2]_{0,\mathrm{LFR}}$ = 130 ppm. At these conditions, $NO_{3\mathrm{exp}}$ decreased from $1.2 \times 10^{14}$ to $2.0 \times 10^{13}$ molecules cm$^{-3}$ s as $RH_{\mathrm{OFR}}$ increased from 11% to 81%. We hypothesize that this result is due to more efficient hydrolysis of $N_2O_5$ to $HNO_3$ on the wetted walls of the OFR at higher RH, thereby suppressing $NO_{3\mathrm{exp}}$ relative to values obtained at lower RH conditions. In an attempt to model this behavior, $k_{\mathrm{w},N_2O_5}$ values input to the model were adjusted as a function of $RH_{\mathrm{OFR}}$. Figure 4 suggests that humidity-dependent $k_{\mathrm{w},N_2O_5}$ values ranging from 0.01 to 0.08 s$^{-1}$ were required to cover the range of measured $NO_{3\mathrm{exp}}$. These values agreed within a factor of 2 or better




with humidity-dependent $k_{w,N_2O_5}$ values ranging from 0.014 to 0.040 s$^{-1}$ measured by Palm et al. (2017) in a similar OFR and were applied in subsequent model calculations.

Figure 5 shows NO$_{3exp}$ as a function of $[O_3]_{0,LFR}$ for measurements with $[NO_2]_{0,LFR}:[O_3]_{0,LFR}$ = 0.5 ± 0.1 and RH$_{OFR}$ = 11 ± 2%. The equivalent ambient photochemical age shown on the right y-axis was calculated assuming a 14-hour average nighttime NO$_3$ mixing ratio of 30 ppt and a 10-hour daytime NO$_3$ mixing ratio of 0 ppt (Asaf et al., 2010). NO$_{3exp}$ increased with increasing $[O_3]_{0,LFR}$ due to increased NO$_3$ production from higher $[N_2O_5]$. Over the range of measured conditions, increasing $[O_3]_{0,LFR}$ from 33 to 7092 ppm increased NO$_{3exp}$ from 6.4×10$^{12}$ to 4.0×10$^{15}$ molec cm$^{-3}$ s$^{-1}$. The black line in Figure 5 represents NO$_{3exp}$ modeled using the mechanism shown in Table S2. Measured and modeled NO$_{3exp}$ values agreed within a factor of 2 or better above $[O_3]_{0,LFR} \approx 40$ ppm, and the gain in NO$_{3exp}$ as a function of $[O_3]_{0,LFR}$ was highest between $[O_3]_{0,LFR} \approx 10$ and 300 ppm. Over this range of $[O_3]_{0,LFR}$, the NO$_2$ oxidation lifetime with respect to O$_3$ decreased from 115 s to 4 s. Because $\tau_{LFR}$ = 20 s, in this range of LFR conditions, the NO$_2$ lifetime in the LFR was long enough that high NO$_2$ levels exiting the LFR suppressed NO$_{3exp}$ in the OFR. On the other hand, increasing $[O_3]_{0,LFR}$ from 300 to 7000 ppm decreased the NO$_2$ oxidation lifetime with respect to O$_3$ from 4 s to 0.2 s, and $[NO_2]$ exiting the LFR was too low to significantly affect NO$_{3exp}$. To support this hypothesis, Figure 6 plots NO$_{3exp}$ as a function of $[NO_2]_{0,LFR}:[O_3]_{0,LFR}$ at $[O_3]_{0,LFR}$ = 250 ± 20 ppm and 6850 ± 400 ppm. Here, we incorporated NO$_{3exp}$ values obtained over RH$_{OFR}$= 11% to 81% for better statistics, and normalized each NO$_{3exp}$ value to the maximum NO$_{3exp}$ obtained at the same RH. Figure 6 shows that at $[O_3]_{0,LFR}$= 250 ppm, maximum NO$_{3,exp}$ was achieved at $[NO_2]_{0,LFR}:[O_3]_{0,LFR} \approx 0.5$ to 0.7. On the other hand, at $[O_3]_{0,LFR}$ = 6850 ppm, maximum NO$_{3exp}$ value was achieved at $[NO_2]_{0,LFR}:[O_3]_{0,LFR} \approx 1.2$.

In a related set of experiments, IBBCEAS measurements of the NO$_2$:NO$_3$ ratio at the exit of the OFR (obtained from Figure 3a spectra) confirmed that significantly higher NO$_2$ levels were present in the OFR at higher $[NO_2]_{0,LFR}:[O_3]_{0,LFR}$, as expected. For example, at $[O_3]_{0,LFR}$ = 150 ppm and $[NO_2]_{0,LFR}$ = 112 ppm, NO$_2$:NO$_3$ = 28, whereas at $[O_3]_{0,LFR}$ = 160 ppm and $[NO_2]_{0,LFR}$ = 320 ppm, NO$_2$:NO$_3$ = 613. NO$_2$:NO$_3$, along with NO$_3$:O$_3$ and NO$_2$:NO$_3$, has important implications for the fate of organic species in OFR-iN$_2$O$_5$ that are discussed in the following sections.

## 3.4 Model characterization of OFR-iN$_2$O$_5$: NO$_3$:O$_3$, NO$_2$:NO$_3$, and NO$_2$:O$_2$

To examine OFR-iN$_2$O$_5$ performance over a wider range of conditions, Figure 7 plots the mean NO$_{3exp}$ , [O$_3$], NO$_3$:O$_3$, NO$_2$:NO$_3$, and NO$_2$:O$_2$ values obtained with the model as a function of $[O_3]_{0,LFR}$ = 10 ppm to 10$^5$ ppm (10%), for $[NO_2]_{0,LFR}:[O_3]_{0,LFR}$ = 0.01, 0.1, 0.5, 1.0, 1.5, 1.8 and 2.0. Three observations are apparent from Figure 7. First, at $[O_3]_{0,LFR}$ < 1000 ppm and $[NO_2]_{0,LFR}:[O_3]_{0,LFR}$ = 0.1 to 1.8, maximum NO$_{3exp}$ increased with decreasing $[NO_2]_{0,LFR}:[O_3]_{0,LFR}$ (Fig. 7a). Above $[O_3]_{0,LFR} \approx 2000$ ppm, NO$_{3exp}$ was less sensitive to $[NO_2]_{0,LFR}:[O_3]_{0,LFR}$. Second, maximum NO$_3$:O$_3$ increased with increasing $[NO_2]_{0,LFR}:[O_3]_{0,LFR}$ (Figure 7c). Third, the $[NO_2]_{0,LFR}:[O_3]_{0,LFR}$ = 2.0 case demonstrated unique behavior relative to the other cases because residual O$_3$ exiting the LFR was low (< 10 ppm) because of nearly complete conversion of O$_3$ to N$_2$O$_5$ inside the LFR (Figure 7b). Consequently, the high residual $[NO_2]$ suppressed NO$_{3exp}$ by one to two orders of magnitude relative to $[NO_2]_{0,LFR}:[O_3]_{0,LFR}$ < 2 cases (Fig. 7a) and generated enhanced NO$_3$:O$_3$, NO$_2$:NO$_3$, and NO$_2$:O$_2$ values. In addition, NO$_2$:NO$_3$ ratios obtained from IBBCEAS measurements at $[O_3]_{0,LFR}$ = 150 to 160 ppm



and $[NO_2]_{0,LFR}$:$[O_3]_{0,LFR}$ = 0.75, 1.0 and 2.0 are shown in Figure 7d. The measured $NO_2$:$NO_3$ values are comparable to, or lower than, the modeled $NO_2$:$NO_3$ values obtained at similar conditions, and therefore broadly support using model results to further investigate the fate of (1) $RO_2$ that are formed from $NO_3$ oxidation of VOCs, (2) alkyl radicals that are reactive towards $NO_2$ and $O_2$, and (3) VOCs that are reactive towards $O_3$ and $NO_3$ in the following sections.

### 3.4.1  Fate of organic peroxy radicals ($RO_2$) formed from $NO_3$ + VOC reactions

$RO_2$ react with NO, $NO_2$, $NO_3$, $HO_2$, or other $RO_2$ to generate alkoxy (RO) radicals, peroxynitrates ($RO_2NO_2$), hydroperoxides or organic peroxides, and may additionally undergo autooxidation via sequential isomerization and $O_2$ addition. To investigate the fate of $RO_2$ as a function of OFR-i$N_2O_5$ conditions, we applied the methodology of Peng et al. (2019) by calculating the fractional oxidative loss of a generic alkyl or acyl $RO_2$ to each of these species over the range of conditions shown in Figure 7. Kinetic data from Orlando and Tyndall (2012) that was used in these calculations is summarized in Table S3. Under almost all OFR-i$N_2O_5$ conditions shown in Figure 7, $RO_2$ reactions with NO, $HO_2$, and $RO_2$ were minor ($< 1\%$) loss pathways compared to reaction with $NO_2$ and $NO_3$, although the $RO_2$ + $RO_2$ pathway is potentially more important at higher $NO_3R_{ext}$. To investigate the relative importance of competing $RO_2$ + $NO_2$ and $RO_2$ + $NO_3$ pathways, we defined the fractional reactive loss of $RO_2$ due to $NO_3$, $F_{RO2+NO3}$:

$$F_{RO2+NO3} = \frac{k_{NO_3}[NO_3]}{k_{NO_3}[NO_3] + k_{NO_2}[NO_2]} \tag{2}$$

Figures 8a and 8b show $F_{RO2+NO3}$ calculated for alkyl and acyl $RO_2$ respectively. To simplify the analysis, we assumed that the thermal decomposition of $RO_2NO_2$ species formed from $RO_2$ + $NO_2$ reactions was slow compared to $\tau_{OFR}$. This assumption generates a lower limit $F_{RO2+NO3}$ value for the alkyl $RO_2$ case, where $RO_2NO_2$ decomposition occurs on timescales of seconds or less (Orlando and Tyndall, 2012)), but has minimal influence on the acyl-$RO_2$ case due to higher thermal stability of peroxyl acyl nitrates. For alkyl $RO_2$, Figure 8a shows that $F_{RO2+NO3}$ = 0.5 was achieved between $[NO_2, O_3]_{0,LFR}$ = (125 ppm, 250 ppm) and (3240 ppm, 1800 ppm). For acyl $RO_2$, due to faster reaction with $NO_2$, Figure 8b shows that $F_{RO2+NO3}$ = 0.5 was achieved using $[NO_2, O_3]_{0,LFR}$ = (350 ppm, 700 ppm) to (1.1%, 0.6%).

To investigate the feasibility of generating OFR-i$N_2O_5$ conditions where $RO_2$ loss is dominated by autooxidation, we calculated the lifetime of alkyl and acyl $RO_2$ ($\tau_{RO_2}$) over the range of OFR-i$N_2O_5$ conditions shown in Figures 7 and 8a-b. As shown in Figures 8d-e, maximum $\tau_{RO_2} \approx 1.4$ s (alkyl) and 0.4 s (acyl) were obtained at $[NO_2]_{0,LFR} \approx 2$ ppm and $[O_3]_{0,LFR} \approx 200$ ppm. At lower $[O_3]_{0,LFR}$, $\tau_{RO_2}$ decreased due to faster $RO_2$ + $NO_2$ reaction rate, and at higher $[O_3]_{0,LFR}$, $\tau_{RO_2}$ decreased due to faster $RO_2$ + $NO_3$ reaction rate. Because $RO_2$ autooxidation timescales range from 0.005 to 200 s depending on the specific $RO_2$ composition (Crounse et al., 2013), OFR-i$N_2O_5$ may achieve autooxidation-dominant conditions for some $RO_2$ but not for others.

### 3.4.2  Fate of aromatic alkyl radicals (R) formed from $NO_3$ + VOC reactions

The majority of $R$ that are generated from $NO_3$ oxidation of VOCs quickly react with $O_2$ to generate $RO_2$. However, $NO_3$ oxidation of a subset of aromatic VOCs generates $R$ that react more slowly with $O_2$, thereby enabling competing reactions





with $NO_2$. For example, the phenoxy radical ($C_6H_5O$) generated from $NO_3$ oxidation of phenol ($C_6H_5OH$) has $k_{O_2}:k_{NO_2} < 2.4\times10^{-9}$ (Platz et al., 1998), and the $C_{10}H_7NO_3$ radical that is generated from $NO_3$ oxidation of naphthalene ($C_{10}H_8$) has $k_{O_2}:k_{NO_2} < 4\times10^{-7}$ (Atkinson et al., 1994). Alkyl radicals generated from $NO_3$ oxidation of other PAH may behave similarly to $C_{10}H_7NO_3$ but kinetic data are unavailable in the literature. To investigate the relative importance of competing $R + NO_2$ 5 and $R + O_2$ reactions in these systems, we defined the fractional reactive loss of $R$ with respect to $O_2$, $F_{R+O_2}$:

$$F_{R+O_2} = \frac{k_{O_2}[O_2]}{k_{O_2}[O_2] + k_{NO_2}[NO_2]} \tag{3}$$

Figure 8c shows $F_{R+O_2}$ over the same OFR-i$N_2O_5$ operating conditions used to generate Figures 7 and 8a-b. For $C_6H_5O$ (not shown), $F_{R+O_2} < 0.08$ over the entire range of OFR-i$N_2O_5$ conditions shown in Figures 7e and 8c. For $C_{10}H_7NO_3$, $F_{R+O_2} \geq 0.5$ was achieved for the majority of OFR-i$N_2O_5$ conditions where $[NO_2]_{0,LFR}:[O_3]_{0,LFR} \leq 0.1$, and also between 10 $[NO_2, O_3]_{0,LFR} = (100\ ppm, 200\ ppm)$ and $(5000\ ppm, 10000\ ppm)$. The use of $[NO_2]_{0,LFR}:[O_3]_{0,LFR} \geq 1$ always generated conditions where the reaction rate of $R + NO_2$ exceeded $R + O_2$.

### 3.4.3   Fate of VOCs reactive towards $O_3$ and $NO_3$

We defined the fractional reactive loss of a VOC with respect to $NO_3$, $F_{VOC+NO_3}$:

$$F_{VOC+NO_3} = \frac{k_{NO_3}[NO_3]}{k_{NO_3}[NO_3] + k_{O_3}[O_3]} \tag{4}$$

15 and established $F_{VOC+NO_3} = 0.9$ as the criterion for $NO_3$-dominated oxidative loss. Figure 9 plots $NO_3:O_3$ at which $F_{VOC+NO_3}$ = 0.9 for several classes of organic compounds with published $k_{NO_3}$ and $k_{O_3}$ values greater than $10^{-16}$ and $10^{-19}$ cm$^{-3}$ molecules$^{-1}$ s$^{-1}$, respectively. This figure therefore excludes compounds such as alkanes and monocyclic aromatics that react slowly with $NO_3$ and are essentially unreactive towards $O_3$ ($F_{NO_3} \approx 1$). $NO_3:O_3$ values that correspond to $[NO_2]_{0,LFR}$ and $[O_3]_{0,LFR} = [2\ ppm, 200\ ppm]$, $[150\ ppm, 300\ ppm]$, and $[5400\ ppm, 3000\ ppm]$ are represented by horizontal bands with 20 upper and lower limit values calculated assuming $k_{w,N_2O_5}$ values of 0.01 and 0.08 s$^{-1}$ (Section 3.3). These LFR inputs generated OFR-i$N_2O_5$ conditions that maximize $RO_2$ lifetime and $NO_3:O_3$ at $[NO_2]:[O_3]_{0,LFR} = 0.5$ and 1.8, respectively (Figures 7-8). Figures 7 and 9 together with kinetic data in the literature suggest that injection of 2 ppm $NO_2$ and 200 ppm $O_3$ into the LFR was sufficient to achieve $F_{VOC+NO_3} \geq 0.9$ for phenols, PAHs with no double bonds, and mono- and sesquiterpenes with 1 double bond at low $RH_{OFR}$. Increasing $[NO_2]_{0,LFR}$ to 150 ppm and $[O_3]_{0,LFR}$ to 300 ppm additionally achieved $F_{VOC+NO_3}$ 25 $\geq 0.9$ for acenaphthylene, isoprene, and mono- and sesquiterpenes with 1 double bond at elevated $RH_{OFR}$. Further increasing $[NO_2]_{0,LFR}$ to 5400 ppm and $[O_3]_{0,LFR}$ to 3000 ppm achieved $F_{VOC+NO_3} \geq 0.9$ for $\geq$ C3 linear alkenes, unsaturated aldehydes, and mono- and sesquiterpenes with 2 double bonds at low $RH_{OFR}$. While $[NO_2, O_3] = [20\%, 10\%]$ (not shown) achieved $F_{VOC+NO_3} \geq 0.9$ for (E)-3-penten-2-one and ethene, the corresponding $NO_{3exp} \approx 10^{14}$ molecules cm$^{-3}$ s achieved at this condition (Figure 7a) was insufficient to oxidize more than 1-2% of the initial ethene concentration due to its slow $NO_3$ rate 30 constant (Atkinson, 1991).



## 3.5 NO$_3$ estimation equation for OFR-iN$_2$O$_5$

Previous studies reported empirical OH exposure algebraic estimation equations for use with OFRs (Li et al., 2015; Peng et al., 2015, 2018; Lambe et al., 2019). These equations parameterize OH$_{exp}$ as a function of readily-measured experimental parameters, therefore providing a simpler alternative than detailed photohemical models for experimental planning and analysis. Here,

we expand on those studies by deriving an NO$_{3exp}$ estimation equations for OFR-iN$_2$O$_5$. Model results obtained from the base case of the model – a VOC reacting with NO$_3$ at $2.5 \times 10^{-12}$ cm$^3$ molecule$^{-1}$ s$^{-1}$ as surrogate of NO$_3$R$_{ext}$ – were used to derive the following equation that allows estimating NO$_{3exp}$ for OFR-iN$_2$O$_5$:

$$\log[(\mathrm{NO}_3)_{\mathrm{exp}}] = a + b\log[273.15 + \mathrm{T_{OFR}}] + c\log[\tau_{\mathrm{OFR}}] + d\log[\mathrm{NO}_2]_{0,\mathrm{LFR}} + e\log[\mathrm{O}_3]_{0,\mathrm{LFR}} \cdot \mathrm{T_{OFR}} \quad (5)$$

$$+ f\log[\mathrm{k_{w_{OFR},N_2O_5}}] + \log\left(\frac{[\mathrm{NO}_2]_{0,\mathrm{LFR}}}{[\mathrm{O}_3]_{0,\mathrm{LFR}}}\right) \cdot (g\,(\log[\mathrm{O}_3]_{0,\mathrm{LFR}})^2 + h\log[\mathrm{O}_3]_{0,\mathrm{LFR}}) - \frac{[\mathrm{NO}_2]_{0,\mathrm{LFR}}}{[\mathrm{O}_3]_{0,\mathrm{LFR}}} \cdot (i + j\log[\mathrm{O}_3]_{0,\mathrm{LFR}})$$

$$+ k\log(\mathrm{NO}_3\mathrm{R})_{\mathrm{ext}} + l\log[\mathrm{NO}_2]_{0,\mathrm{LFR}} \cdot \mathrm{T} + m\log[\mathrm{O}_3]_{0,\mathrm{LFR}} \cdot \log\mathrm{k_{w_{OFR},N_2O_5}}$$

The phase space of OFR-iN$_2$O$_5$ parameters for fitting Equation 5 to NO$_{3exp}$ model results was defined as follows: $[\mathrm{O}_3]_{0,\mathrm{LFR}}$ = 10-1000 ppm, $[\mathrm{NO}_2]_{0,\mathrm{LFR}}$ = 10-1000 ppm, $[\mathrm{NO}_2]_{0,\mathrm{LFR}}:[\mathrm{O}_3]_{0,\mathrm{LFR}} \leq 2$, NO$_3$R$_{ext}$ = 1-200 s$^{-1}$, $k_{\mathrm{w_{OFR},N_2O_5}}$ = 0.01-0.08 s$^{-1}$, $T_{\mathrm{OFR}}$ = 0 - 40°C, and $\tau_{\mathrm{OFR}}$ = 60 - 300 s. The cases where $\mathrm{O}_3]_{0,\mathrm{LFR}} > 1000$ ppm and/or $[\mathrm{NO}_2]_{0,\mathrm{LFR}}:[\mathrm{O}_3]_{0,\mathrm{LFR}} > 2$ were not considered because of less practical interest. We explored 11, 11, 7, 4, and 5 logarithmically evenly distributed values

in the ranges of $[\mathrm{O}_3]_{0,\mathrm{LFR}}$, $[\mathrm{NO}_2]_{0,\mathrm{LFR}}$ (11 values over 10–1000 ppm), NO$_3$R$_{ext}$, $k_{\mathrm{w,N_2O_5}}$, and $\tau_{\mathrm{OFR}}$, respectively. Due to significantly different chemical regimes in different parts of the phase space, fit coefficients that are reported in Table 1 were obtained by fitting the same functional form (Equation 5) over 3 sub-phase spaces with the following additional constraints: (i) $[\mathrm{NO}_2]_{0,\mathrm{LFR}}:[\mathrm{O}_3]_{0,\mathrm{LFR}}$ = 0-1 and NO$_3$R$_{ext}$ = 20-200 s$^{-1}$; (ii) $[\mathrm{NO}_2]_{0,\mathrm{LFR}}:[\mathrm{O}_3]_{0,\mathrm{LFR}}$= 0-1 and NO$_3$R$_{ext}$ = 1-20 s$^{-1}$ (iii) $[\mathrm{NO}_2]_{0,\mathrm{LFR}}:[\mathrm{O}_3]_{0,\mathrm{LFR}}$ = 1-2. For these 3 subspaces, 10080, 13440, and 5880 model cases respectively were simulated. In

Equation 5, the terms involving the coefficients $g$–$j$ were included to reproduce the relationship between normalized NO$_{3exp}$ and $[\mathrm{NO}_2]_{0,\mathrm{LFR}}:[\mathrm{O}_3]_{0,\mathrm{LFR}}$ shown in Figure 5. Logarithms of first- and second-order terms were successively added until no further fit quality improvement was achieved. Figure 10 compares NO$_{3exp}$ estimated from Equation 5 and calculated from the model described in Section 2.2. The mean absolute value of the relative deviation was 49% which is comparable with results obtained for previous estimation equations with significant NO$_y$ chemistry (Peng et al., 2018).

## 3.6 SOA generation from $\beta$-pinene + NO$_3$

To apply the OFR-iN$_2$O$_5$ technique to SOA formation studies, we generated SOA from $\beta$-pinene + NO$_3$ in the absence of seed particles using $[\mathrm{O}_3]_{0,\mathrm{LFR}}$ = 300 ppm, $[\mathrm{NO}_2]_{0,\mathrm{LFR}}$ = 150 ppm, and RH$_{\mathrm{OFR}} \approx 1\%$. PTR-MS measurements confirmed complete consumption of $\beta$-pinene, and numerous product ions were detected. The largest ions detected were $(\mathrm{H}^+)\mathrm{C}_9\mathrm{H}_{14}\mathrm{O}$ and $(\mathrm{H}^+)\mathrm{C}_{10}\mathrm{H}_{14}$ which may correspond to nopinone ($\mathrm{C}_9\mathrm{H}_{14}\mathrm{O}$) and fragmentation/decomposition products of $\mathrm{C}_{10}\mathrm{H}_{17}\mathrm{NO}_4$

respectively (Hallquist et al., 1999; Claflin and Ziemann, 2018). The mass yield of SOA ranged from 0.03 to 0.39 over $\beta$-pinene mixing ratios ranging from 20-400 ppbv that were injected into the OFR. These yield values are broadly consistent





with previous environmental chamber studies (Ng et al., 2017) but are lower than chamber SOA yields obtained at the same $\beta$-pinene mixing ratio, presumably due to the absence of seed particles in the OFR (Lambe et al., 2015). To compare results obtained using OFR-iN$_2$O$_5$ with a conventional environmental chamber method, Figures 11a-b show HR-ToF-AMS spectra of SOA generated from NO$_3$ oxidation of $\beta$-pinene in the Georgia Tech chamber (Boyd et al., 2015) and in the OFR, along with a scatter plot of relative ion abundances present in the two spectra (Figure 11c). The same spectra are presented on a logarithmic scale in Figure S4. As is evident, $\beta$-pinene + NO$_3$ SOA generated in the chamber and OFR exhibit a high degree of similarity (linear regression slope = 0.98 and $r^2$ = 0.99). The largest ion signal was observed at NO$^+$, which, along with signal at NO$_2^+$ and NO$^+$: NO$_2^+$ = 6.7, is consistent with the formation of particulate organic nitrates (Farmer et al., 2010). Signals observed at CHO$^+$, C$_2$H$_3$O$^+$, and other C$_x$H$_y$O$_{>1}^+$ ions suggest the presence of other multifunctional oxidation products.

## 4  Conclusions

OFR-iN$_2$O$_5$ complements recently developed methods that enable NO$_x$-dependent photooxidation studies in OFRs such as OFR-iN$_2$O and OFR-iC$_3$H$_7$ONO (Lambe et al., 2017; Peng et al., 2018; Lambe et al., 2019) by enabling studies of nighttime NO$_3$-initiated oxidative aging processes. Important OFR-iN$_2$O$_5$ parameters are [O$_3$], [NO$_2$], [H$_2$O], $T$, NO$_3$R$_{ext}$, and $\tau_{OFR}$. By contrast, important OFR-iN$_2$O and OFR-iC$_3$H$_7$ONO parameters are UV intensity, external OH reactivity (OHR$_{ext}$), $\tau_{OFR}$, and either [O$_3$] + [H$_2$O] + [N$_2$O] or [C$_3$H$_7$ONO]. Notably, NO$_3$R$_{ext}$ is typically less significant in OFR-iN$_2$O$_5$ than OHR$_{ext}$ in OFR-iN$_2$O or OFR-iC$_3$H$_7$ONO because (1) most compounds are less reactive towards NO$_3$ than OH, (2) NO$_{3exp}$ is higher than OH$_{exp}$, and (3) the internal NO$_3$ reactivity of OFR-iN$_2$O$_5$, which is dominated by the NO$_3$ + NO$_2$ reaction, is larger and easier to manipulate than the internal OH reactivity of OFR-iN$_2$O and OFR-iC$_3$H$_7$ONO, which is dominated by OH + HO$_2$ and OH + NO$_2$ reactions. To identify optimal OFR-iN$_2$O$_5$ conditions for different applications, we characterized NO$_{3exp}$, $\tau_{RO_2}$, $F_{RO_2+NO_3}$, $F_{R+O_2}$ and $F_{VOC+NO_3}$ at [O$_3$]$_{0,LFR}$ = 10 ppm to 10%, [NO$_2$]$_{0,LFR}$:[O$_3$]$_{0,LFR}$ = 0.01 to 2.0, and RH$_{OFR}$ = 7 to 85%. Optimal NO$_{3exp}$ was achieved by minimizing [H$_2$O] in the OFR and associated humidity-dependent N$_2$O$_5$ wall losses. This is contrary to most OFR techniques that are used to generate OH radicals, where optimal OH$_{exp}$ is achieved by maximizing [H$_2$O] and associated OH production from the O($^1$D) + H$_2$O reaction and/or H$_2$O photolysis at $\lambda$ = 185 nm.

Figure 12 presents image plots that represent OFR-iN$_2$O$_5$ conditions suitable for generating optimal NO$_{3exp}$, NO$_3$:O$_3$, NO$_2$:NO$_3$, and $\tau_{RO_2}$ values at the lower and upper-limit $k_{w,N_2O_5}$ values that were measured. Most OFR-iN$_2$O$_5$ conditions using [O$_3$]$_{0,LFR}$ > 200 ppm generated NO$_{3exp}$ > 1.5×10$^{12}$ molecules cm$^{-3}$ s (Figures 12a-b), which is sufficient to oxidize isoprene and compounds with similar $k_{NO_3}$; for reference, NO$_{3exp}$ > 1.6×10$^{11}$ molecules cm$^{-3}$ s is required to oxidize $\alpha$-pinene. At [O$_3$]$_{0,LFR}$ >200 ppm and [NO$_2$]$_{0,LFR}$:[O$_3$]$_{0,LFR}$ > 0.5, OFR-iN$_2$O$_5$ generated NO$_3$:O$_3$ > 10$^{-3}$ at $k_{w,N_2O_5}$ = 0.01 s$^{-1}$ (Figure 12c), which achieved $F_{VOC+NO_3}$ > 0.9 for mono- and sesquiterpenes with 1 double bond, most PAHs, and phenol/methoxyphenol species. Achieving NO$_3$:O$_3$ > 10$^{-3}$ at $k_{w,N_2O_5}$ = 0.08 s$^{-1}$ was more challenging (Figure 12d). Increasing [O$_3$]$_{0,LFR}$ decreased [NO$_2$]:[NO$_3$] and therefore increased $F_{RO_2+NO_3}$ (Figures 12e-f). On the other hand, decreasing [O$_3$]$_{0,LFR}$ or increasing $k_{w,N_2O_5}$, and, consequently, NO$_{3exp}$, increased $\tau_{RO_2}$ (Figures 12g-h), potentially allowing more time for autooxidation processes to occur. The best overlap between OFR-iN$_2$O$_5$ conditions that achieved $F_{RO_2+NO_3}$ > 0.9



and $\tau_{RO_2} > 1$ s were obtained with $[NO_2]_{0,LFR} \approx$ 2-3 ppm and $[O_3]_{0,LFR} \approx$ 200-300 ppm. Because atmospheric $NO_2$:$NO_3$ is highly variable and often much larger than $NO_2$:$NO_3$ achieved using OFR-i$N_2O_5$ (Brown et al., 2003; Stutz et al., 2004), simply attempting to maximize $F_{RO_2+NO_3}$ may not always be necessary and has tradeoffs such as decreasing $NO_3$:$O_3$ and $F_{VOC+NO_3}$. OFR-i$N_2O_5$ was more difficult to apply to species such as unsaturated carbonyls and mono- and sequiterpenes

with multiple double bonds that react more efficiently with $O_3$ than other VOCs; here, alternative $NO_3$ generation techniques that do not introduce $O_3$ to the OFR warrant consideration, even though they are more difficult to implement (Palm et al., 2017).

Because OFR-i$N_2O_5$ can continuously generate $N_2O_5$ and $NO_3$ at room temperature, it is significantly easier to apply in continuous flow reactor studies than related techniques. However, in addition to the aforementioned considerations, high $N_2O_5$

and $HNO_3$ concentrations that are generated using OFR-i$N_2O_5$ complicate the application of techniques such as iodide-adduct chemical ionization mass spectrometry due to efficient reactions between the iodide reagent ion and $N_2O_5$ or $HNO_3$ (Lee et al., 2014). Future applications of OFR-i$N_2O_5$ will investigate the $NO_3$-initiated OVOC and SOA formation potential of simple and complex precursors in laboratory and field studies.

*Code and data availability.* Data and KinSim mechanisms presented in this manuscript are available upon request. The KinSim kinetic

solver is freely available at http://tinyurl.com/kinsim-release.

*Author contributions.* AL, EW, and AA conceived and planned the experiments. AL, JK, FM, LW, PC, AA, and JEP carried out the experiments. MC and AF performed the IBBCEAS measurements and data analysis. AL, JJ and ZP conceived and planned the model simulations, and AL and ZP carried out the model simulations. AL, EW, ZP, and JJ contributed to the interpretation of the results. AL took the lead in writing the manuscript. All authors provided feedback on the manuscript.

*Competing interests.* The authors declare no competing interests.

*Acknowledgements.* AL thanks Christopher Boyd and Sally Ng (Georgia Tech) for sharing AMS data obtained in their environmental chamber, and the following colleagues for helpful discussions: Megan Claflin, Manjula Canagaratna, John Jayne, Douglas Worsnop (ARI), William Brune (Pennsylvania State University), Manfred Winnewisser (Ohio State University), Karl Christe (University of Southern California), and Robert Woodward-Massey, Youfeng Wang, and Chunxiang Ye (Peking University). The authors thank the ACMCC and participants of the

ACMCC pON experiment in December 2018, which was supported by the French Ministry of Environment and part of the COST Action CA16109 COLOSSAL and the Aerosol, Clouds, and Trace gases Research InfraStructure (ACTRIS). ZP and JLJ were supported by NSF AGS-1822664 and EPA STAR 83587701-0. This manuscript has not been reviewed by EPA and no endorsement should be inferred.



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



**Table 1.** Fit parameters for $NO_{3exp}$ estimation equation (Equation 5).

| Parameter | Subspace 1 Values | Subspace 2 Values | Subspace 3 Values |
|-----------|-------------------|-------------------|-------------------|
| a | 61.0694 | -59.3835 | 246.416 |
| b | -20.1400 | 27.3434 | -122.229 |
| c | 0.795209 | 0.803508 | 0.581443 |
| d | -0.375825 | 1.18285 | 51.2355 |
| e | 0.0311034 | 0.00815681 | -0.66569 |
| f | 0.888193 | -0.0731138 | -0.0210958 |
| g | -0.379009 | 0.13199 | -0.346062 |
| h | 1.73605 | -0.422009 | -81.9221 |
| i | 0.14737 | 0.035132 | -22.4373 |
| j | 0.261402 | 0.311104 | 13.204 |
| k | -1.22009 | -0.323329 | -0.118988 |
| l | 0.00733645 | -0.004277 | 0.676436 |
| m | -0.957064 | -0.436977 | -0.3983 |

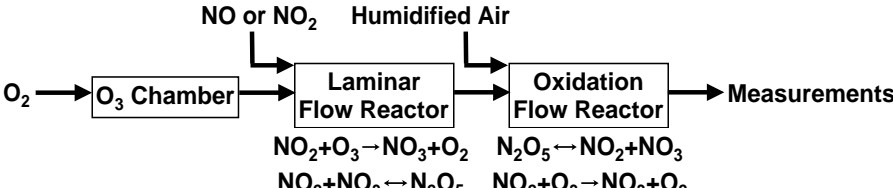

**Figure 1.** Process flow diagram of the OFR-i$N_2O_5$ technique used to generate nitrate radicals ($NO_3$).



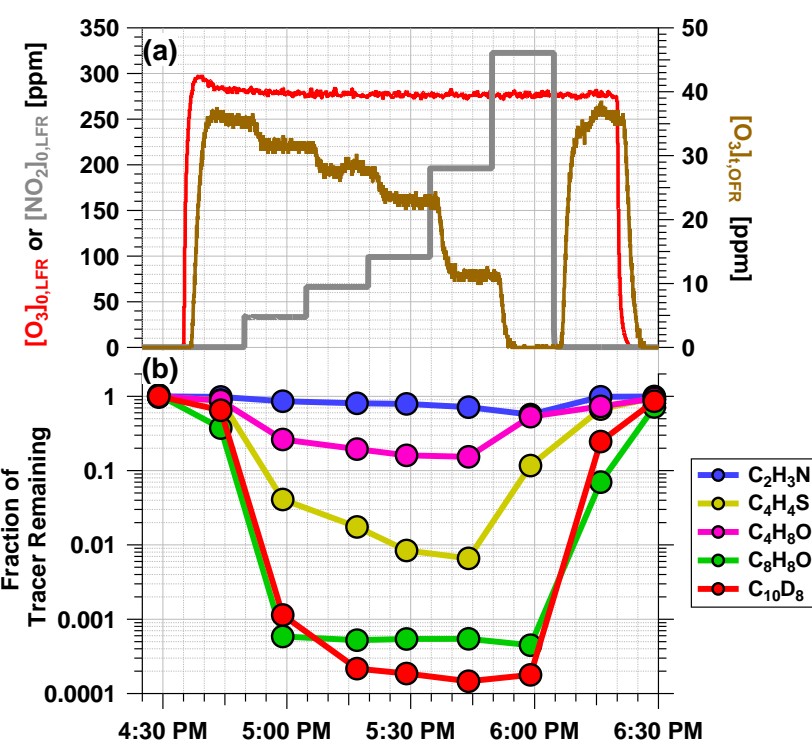

**Figure 2.** Time series from a representative OFR-iN$_2$O$_5$ characterization experiment conducted at RH$_{OFR}$ = 11% of (a) O$_3$ and NO$_2$ mixing ratios input to LFR (left axis) and O$_3$ measured at the exit of the OFR (right axis) (b) VOC tracers measured with PTR-MS: acetonitrile (C$_2$H$_3$N), butanal (C$_4$H$_8$O), thiophene (C$_4$H$_4$S), 2,3-dihydrobenzofuran (C$_8$H$_8$O) and naphthalene-d$_8$ (C$_{10}$D$_8$).



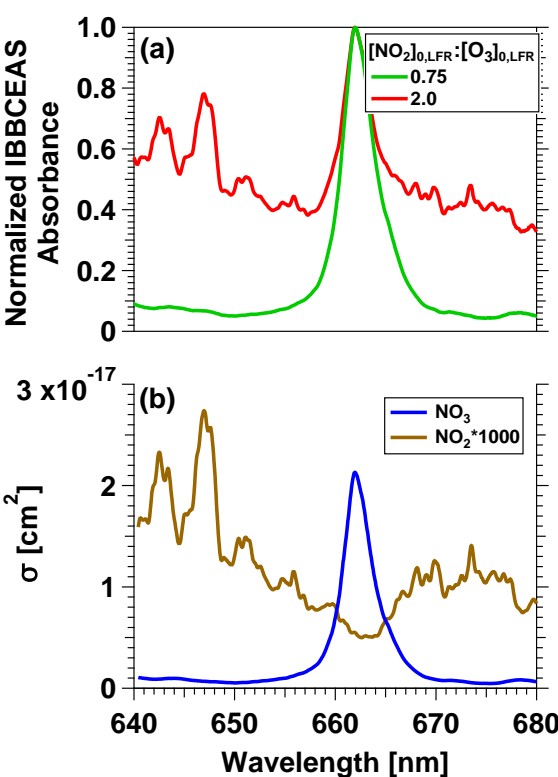

**Figure 3.** (a) IBBCEAS measurements of $NO_2$ and $NO_3$ absorbance obtained from an OFR-i$N_2O_5$ characterization experiment conducted at $[O_3]_{0,LFR}$ = 150-160 ppm and $[NO_2]_{0,LFR}$:$[O_3]_{0,LFR}$ = 0.75 and 2.0. (b) Absorption cross sections of $NO_2$ and $NO_3$ (Vandaele et al., 1998; Orphal et al., 2003).




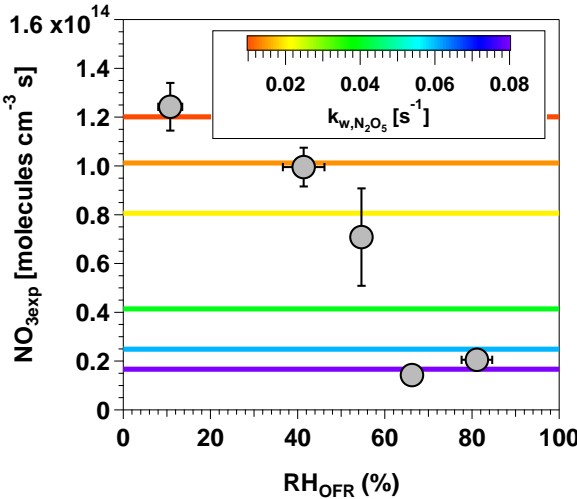

**Figure 4.** $NO_{3exp}$ as a function of $RH_{OFR}$ at $[O_3]_{0,LFR}$ = 250 ppm and $[NO_2]_{0,LFR}$ = 130 ppm. Horizontal lines represent $N_2O_5$ wall loss rate constants ranging from 0.01 to 0.08 $s^{-1}$ that were input to the OFR-i$N_2O_5$ KinSim mechanism (Table S2).

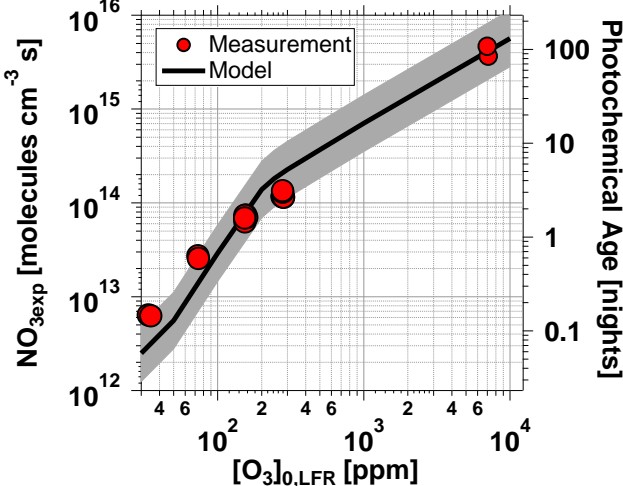

**Figure 5.** $NO_{3exp}$ as a function of $[O_3]_{0,LFR}$ for measurements with $[NO_2]_{0,LFR}:[O_3]_{0,LFR}$ = 0.5 ± 0.1. Equivalent ambient photochemical age was calculated assuming a 14-hour average nighttime $NO_3$ mixing ratio of 30 ppt and 10-hour daytime average $NO_3$ mixing ratio of 0 ppt (Asaf et al., 2010). Model inputs: $k_{w,N_2O_5}$ = 0.01 $s^{-1}$ and $NO_3R_{ext}$ = 0.07 $s^{-1}$ ( $[O_3]_{0,LFR}$ < 1000 ppm) or 0.38 $s^{-1}$ ( $[O_3]_{0,LFR}$ > 1000 ppm). Shaded region encompasses model output scaled by factors of 0.5 and 2.

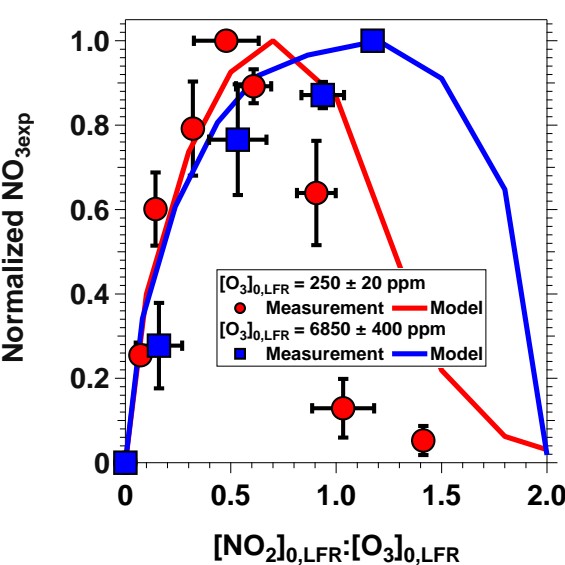

**Figure 6.** $NO_{3exp}$ as a function of $[NO_2]_{0,LFR}:[O_3]_{0,LFR}$ at fixed $[O_3]_{0,LFR}$ values of $250 \pm 20$ and $6850 \pm 400$ ppm and $RH_{OFR} = 11\%$ to 81%. $NO_{3exp}$ values were normalized to maximum $NO_{3exp}$ value obtained at the same RH.



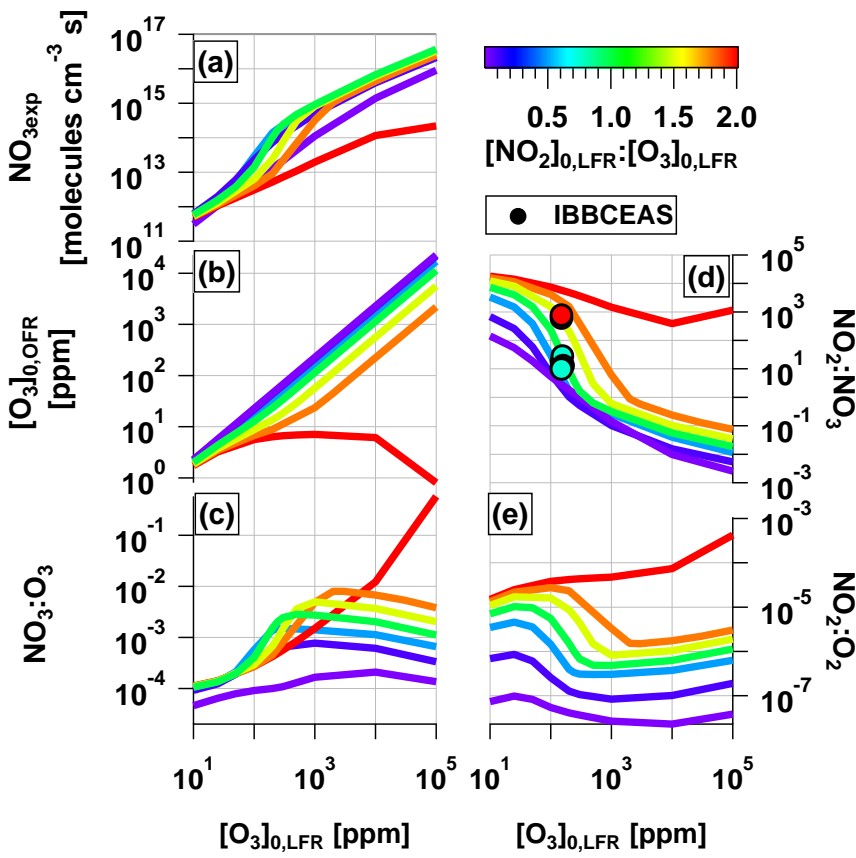

**Figure 7.** Modeled (a) $NO_{3exp}$, (b) $[O_3]$, (c) $NO_3:O_3$, (d) $NO_2:NO_3$, and (e) $NO_2:O_2$ as a function of $[O_3]_{0,LFR} = 10$ ppm to $10^5$ ppm, for $[NO_2]_{0,LFR}:[O_3]_{0,LFR} = 0.01, 0.1, 0.5, 1.0, 1.5, 1.8$ and $2.0$. Model inputs: $k_{w,N_2O_5} = 0.01$ s$^{-1}$, $NO_3R_{ext} = 0.07$ s$^{-1}$. IBBCEAS-measurd $NO_2:NO_3$ values are plotted in (d).



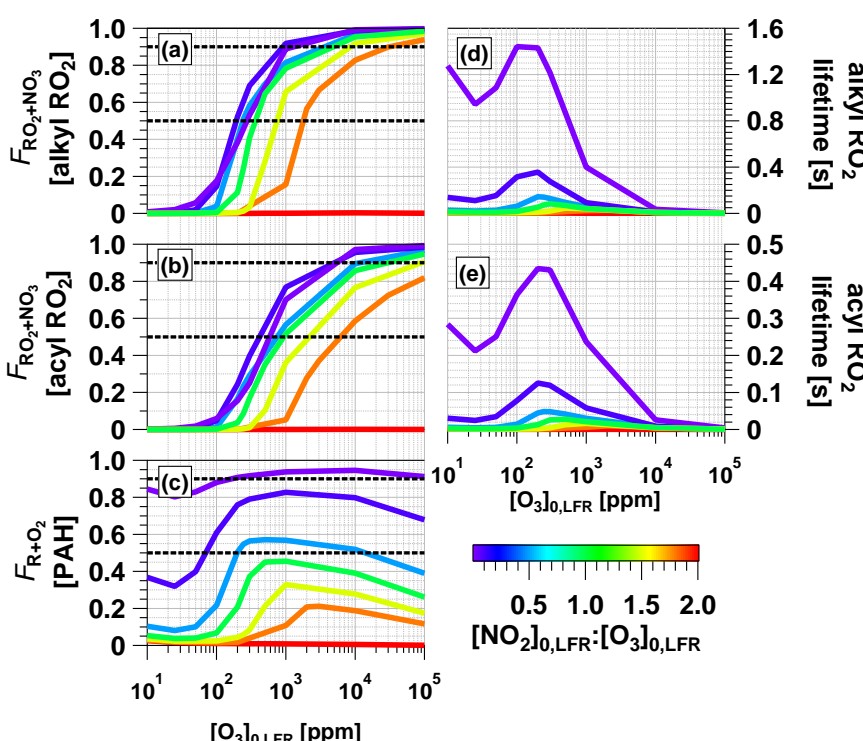

**Figure 8.** $F_{RO_2+NO_3}$ for (a) alkyl and (b) acyl $RO_2$, and (c) $F_{R+O_2}$ over the same OFR-iN$_2$O$_5$ operating conditions and model inputs used to generate Figure 7, with corresponding lifetimes for (d) alkyl and (e) acyl $RO_2$.





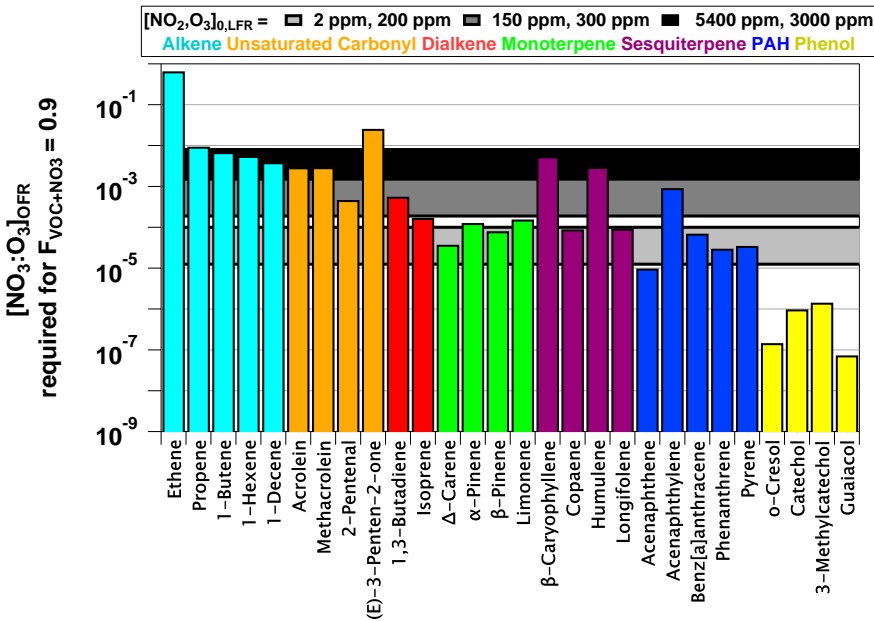

**Figure 9.** $NO_3:O_3$ at which $F_{VOC+NO_3} = 0.9$ for representative VOCs with $k_{NO_3} > 10^{-16}$ and $k_{O_3} > 10^{-19}$ cm$^3$ molecules$^{-1}$ s$^{-1}$ (Manion et al., 2015). Horizontal bands represent upper and lower limit values calculated assuming $k_{w,N_2O_5} = 0.01$ and $0.08$ s$^{-1}$.

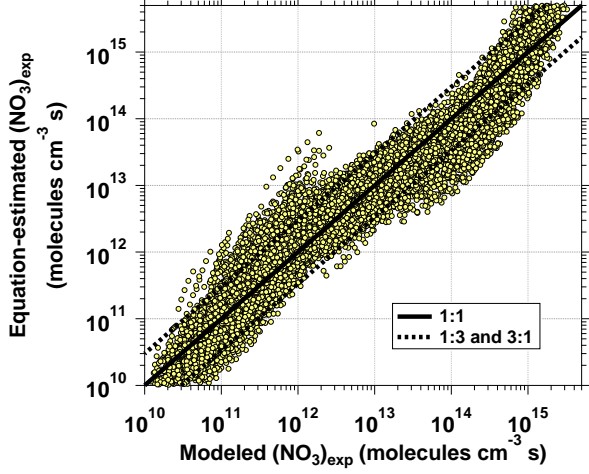

**Figure 10.** $NO_{3exp}$ calculated from estimation equation (Equation 5 and Table 1) as a function of $NO_{3exp}$ calculated from full OFR-i$N_2O_5$ KinSim mechanism (Table S2). Solid and dashed lines correspond to 1:1, 1:3 and 3:1 lines respectively.





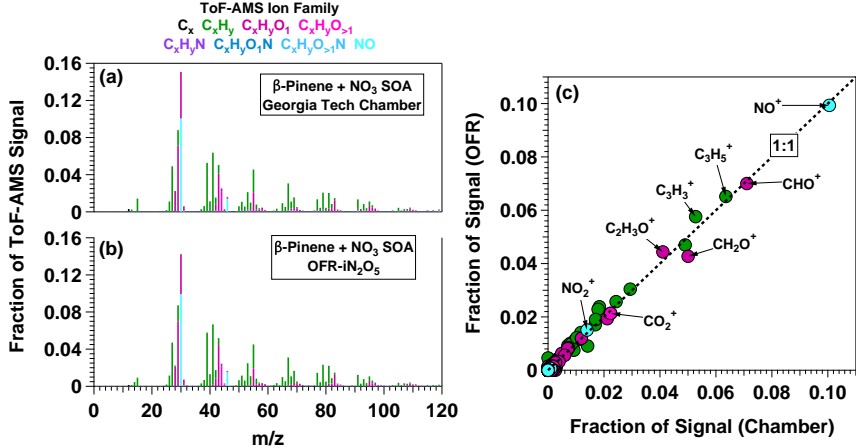

**Figure 11.** AMS spectra of SOA generated from $NO_3$ oxidation of $\beta$-pinene in (a) Georgia Tech environmental chamber (Boyd et al., 2015) and (b) OFR-i$N_2O_5$. Scatter plot in (c) shows spectra generated in the OFR and chamber plotted against each other.

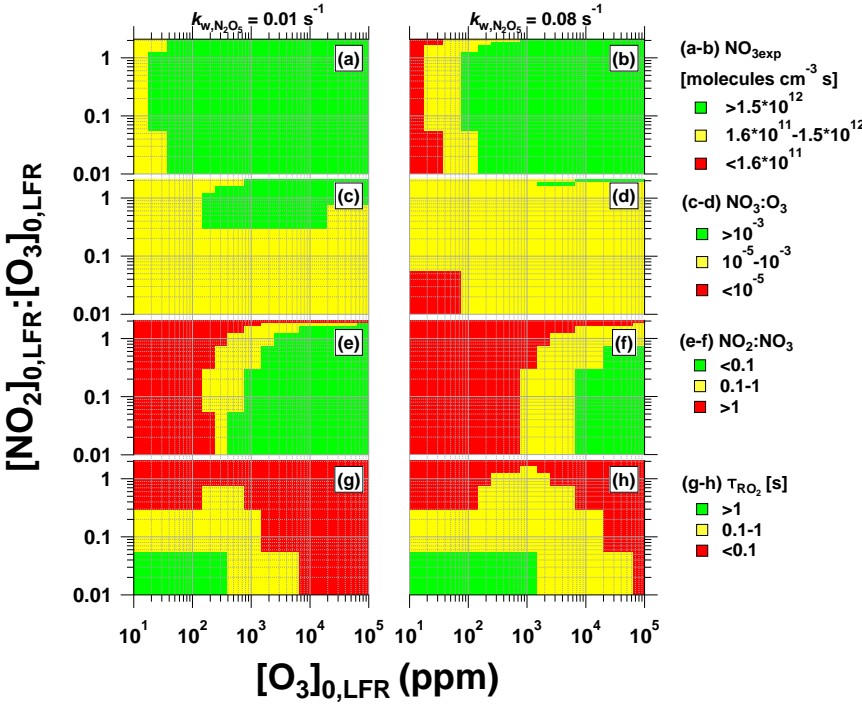

**Figure 12.** Summary of OFR-i$N_2O_5$ operating conditions suitable for maximum (a-b) $NO_3$exp, (c-d) $NO_3$:$O_3$, (e-f) $NO_2$:$NO_3$, and (g-h) $\tau_{RO_2}$ assuming $k_{w,N_2O_5}$ = 0.01 and 0.08 s$^{-1}$.