# Peer review of "Nitrate radical generation via continuous generation of dinitrogen pentoxide in a laminar flow reactor coupled to an oxidation flow reactor"

_Atmospheric Measurement Techniques, 2019_

## Referee Comment (RC1) · Anonymous Referee #1 · 19 Feb 2020

Oxidation flow reactors are widely used in the study of formation and aging of organic aerosols, especially simulating OH radical dominated daytime oxidation process. In this study, the authors extended the use of OFR for studying $NO_3$ radical initialed nighttime oxidation and the aging process by online producing $N_2O_5$ as the source of $NO_3$. By model simulation and measurements, the authors investigated the controlling factors of $NO_3$ exposure, $NO_3$:$O_3$, $NO_2$:$NO_3$ ratios and provided guidelines for experimental design. I believe it will help researchers in using OFR for nighttime chemistry studies. I recommend to accept it after minor revision.

General comments

Regarding the $NO_3$ estimation equation for the ORF-i$N_2O_5$, I wonder how would multiple generation oxidations influence the estimation of $NO_3$ exposure. For example, $NO_3$ radical oxidation of typical BVOCs (isoprene, monoterpenes, sesquiterpenes) produces carbonyls and even products with carbon double bonds. These products are highly reactive toward $NO_3$ radicals which may affect the $NO_3$ exposure estimation. However, these are not considered in the KimSim simulations. Secondly, $NO_3$ oxidation of BVOCs has high SOA yields. I wonder how the uptake of $NO_3$ and $N_2O_5$ by the produced particles affect the simulations.

Specific comments:

(1) BBCES measuring the $NO_3$. 1) The author stated that "$I(\lambda)$ and $I_0(\lambda)$ were the measured transmitted intensities in the presence and absence of $NO_3$". How was the "absence of $NO_3$" achieved? 2) The equation grading the calculation of $\alpha(\lambda)$ is not correct. The $\alpha(\lambda)$ in the cavity also contributed by the bath gas beyond the $NO_3$ radicals. 3) The $NO_3$ radicals are highly reactive and can easily lose to the walls. What is the transmission efficiency of $NO_3$ from the OFR to the cavity? 4) Due to different loss rates of $NO_3$ and $N_2O_5$ to the wall, the equilibrium of $NO_3$ and $N_2O_5$ may change. How good is the measured $NO_3$ concentration in the CRD represent the $NO_3$ radical concentration in the reactor?

 2) The description of the results in section 3.4 is not consistent with the results in the figure.

"First, at $[O_3]_{0,LFR} < 1000$ ppm and $[NO_2]0,LFR:[O_3]_{0,LFR} = 0.1$ to 1.8, maximum $NO_{3exp}$ increased with decreasing $[NO_2]_{0,LFR}:[O_3]_{0,LFR}$ (Fig. 7a)." It is very hard to see the results in the figure when $[O_3]_{0,LFR} < 100$ ppm. For me, it looks like maximum $NO_{3exp}$ first increase with increasing $[NO_2]_{0,LFR}:[O_3]_{0,LFR}$ ratio and then decrease with it, especially when $[O_3]_{0,LFR}$ was in the range of 100-1000 ppm

"Above $[O_3]_{0,LFR} \approx 2000$ ppm, $NO_{3exp}$ was less sensitive to $[NO_2]_{0,LFR}:[O_3]_{0,LFR}$.". This is true except for $[NO_2]_{0,LFR}:[O_3]_{0,LFR}=2.0$

"Second, maximum $NO_3:O_3$ increased with increasing $[NO_2]_{0,LFR}:[O_3]_{0,LFR}$ (Figure 7c)." This statement is true only when the $O_3$ was above 1000 ppm, even get rid of the results from $[NO_2]_{0,LFR}:[O_3]_{0,LFR}=2.0$.

"conversion of $O_3$ to $N_2O_5$ inside the LFR" I fell more comfortable to say "conversion of $O_3$ to $O_2$ inside the LFR".

3) The authors tried to investigate the $RO_2$ fate and considered "$RO_2$ react with NO, $NO_2$, $NO_3$, $HO_2$, or other $RO_2$ to generate alkoxy (RO) radicals, peroxynitrates (RO2NO2), hydroperoxides or organic peroxides, and may additionally undergo autooxidation via sequential isomerization and $O_2$ addition." Recent studies by Berndt et al. (2018 ) revealed that self- and cross-reaction of $RO_2$ radicals would produce dimers effectively. How could this process affect the fate of the $RO_2$ radical?

1. Berndt, T.; Mender, B.; Scholz, W.; Fischer, L.; Herrmann, H.; Kulmala, M.; Hansel, A., Accretion Product Formation from Ozonolysis and OH Radical Reaction of alpha-Pinene: Mechanistic Insight and the Influence of Isoprene and Ethylene. Environ. Sci. Technol. 2018, 52, (19), 11069-11077.

2. Berndt, T.; Scholz, W.; Mentler, B.; Fischer, L.; Herrmann, H.; Kulmala, M.; Hansel, A., Accretion Product Formation from Self- and Cross-Reactions of RO2 Radicals in the Atmosphere. Angew Chem Int Edit 2018, 57, [1], 3820-3824.

---

## Referee Comment (RC2) · Anonymous Referee #3 · 20 Feb 2020

The manuscript "Nitrate radical generation via continuous generation of dinitrogen pentoxide in a laminar flow reactor coupled to an oxidation flow reactor" by Lambe et al. presents a $NO_3$ oxidation flow reactors system for studying the reaction of organic compound with $NO_3$ reactions and secondary organic aerosol from $NO_3$ chemistry. The authors give a very comprehensive characterization of the system by lab study as well as the model simulation. The paper is well written and the results are presented in a clear way. The analysis are sound and the authors proposed ideas will be of interest and helpful to the community. I recommend this paper for publication subjects to some minor comments.

General comments.

1. Section 2.2.1, the section cited some references about the wall loss of $NO_3$ and $N_2O_5$ in Teflon/Pyrex tube, and the $NO_3$ and $N_2O_5$ wall loss in LFR and OFR is extrapolating or interpolation based on the reported results, which weak the results as the wall loss of $NO_3$ and $N_2O_5$ in the system is an important source of uncertainty. The lab quantification of the wall loss in LFR and OFR in the future works can further improve the value of this study. Additionally, page 5, line 9-10, the fixed condition of OFR is ambiguous, please clear it, at least add the simulation results in SI. Page 5, line 11-13, the Extrapolating results is confuse, the reference said 0.04 and 0.009 $s^{-1}$ corresponding to ID (4 and 7 cm), what is corresponding parameter of the $k_{wall,LFR}$ of 0.07 and 0.03 $s^{-1}$ mentioned here?

Specific comments.

2. Page 2, line 29-34, this introduction of the LFR is confusing. The authors can use a schematic figure to show more details about the OFR-i$N_2O_5$ (rather than Figure 1 from references), which would increase the paper's readability.

3. Page 4, line 8. Romanini et al. (1997) is not the right reference of the IBBCEAS principle, I suggest the author replace it by e.g., Fiedler et al., 2003.(Fiedler, S. E., Hese, A., and Ruth, A. A.: Incoherent broad-band cavity-enhanced absorption spectroscopy, Chem Phys Lett, 371, 284-294, 2003.)

4. In Eq .2 the $k_{NO2}$ or $k_{NO3}$ should revised to $k_{NO2+RO2}$ or $k_{NO3+RO2}$, the similar change also applied in Eq. 3 and Eq. 4

---

## Author Response (AR1)

**Response to reviewers for the paper "Nitrate radical generation via continuous generation of dinitrogen pentoxide in a laminar flow reactor coupled to an oxidation flow reactor."**

We thank the referees for their comments on our paper. To guide the review process, we have copied the referee's comments in black text. Our responses are in blue text. Please note that three referees provided quick reports on the discussion manuscript, but Referee #2 did not provide a full review of the discussion manuscript. We respond to Referee #1 and #3 comments, with alterations to the paper indicated **in bold text** below and in annotations to the revised manuscript.

**Anonymous Referee #1**

General comments

1. Regarding the $NO_3$ estimation equation for the OFR-i$N_2O_5$, I wonder how would multiple generation oxidations influence the estimation of $NO_3$ exposure. For example, $NO_3$ radical oxidation of typical BVOCs (isoprene, monoterpenes, sesquiterpenes) produces carbonyls and even products with carbon double bonds. These products are highly reactive toward $NO_3$ radicals which may affect the $NO_3$ exposure estimation. However, these are not considered in the KimSim simulations. Secondly, $NO_3$ oxidation of BVOCs has high SOA yields. I wonder how the uptake of NO3 and N2O5 by the produced particles affect the simulations.

We added the following text to the end of Section 3.5:

P10, L24: "$NO_3R)_{ext}$ of a system will change over the course of multiple generations of $NO_3$ oxidation due to changes in kinetic rate coefficients between different species and $NO_3$ ($k_{NO3}$). The sensitivity of Eq. 5 to changes in ($NO_3R)_{ext}$ depends in part on the relative magnitudes of ($NO_3R)_{ext}$ and the internal $NO_3$ reactivity, ($NO_3R)_{int}$, which is approximately equal to $k_{NO2+NO3}[NO_2]$. If ($NO_3R)_{int} >> (NO_3R)_{ext}$, changes in ($NO_3R)_{ext}$ would have minimal influence on Eq. 5.

In one case study, we examined changes in ($NO_3R)_{ext}$ following conversion of biogenic VOCs (BVOCs) to gas-phase carbonyl oxidation products with known $k_{NO3}$ values. Table S5 compares $k_{NO3}$ of isoprene to methyl vinyl ketone and methacrolein, $\alpha$-pinene to pinonaldehyde, sabinene to sabinaketone, and 3-carene to caronaldehyde. In the limit where 100% of each BVOC is converted to its carbonyl oxidation product(s), ($NO_3R)_{ext}$ decreases by a factor of 200 or greater. Unsaturated organic nitrates that are generated from BVOC + $NO_3$ may also be reactive towards $NO_3$, but $k_{NO3}$ for these species are not available.

In another case study, we examined changes in ($NO_3R)_{ext}$ following conversion of BVOCs to SOA. An effective $k_{NO3}$ for SOA was calculated using the following equation adapted from Lambe et al. (2009):

$$k_{NO_3} = \frac{3}{2} \frac{\gamma \times \bar{c} \times M_{SOA} \times F_{diff}}{D_p \times \rho_p \times N_A}$$

where $F_{diff}$ is a correction factor accounting for diffusion limitations to the particle surface in the transition regime (Fuchs and Sutugin, 1970):

$$F_{diff} = \frac{1 + 6 \times \frac{D_{NO_3}}{\bar{c} \times D_p}}{1 + 10.26 \times \frac{D_{NO_3}}{\bar{c} \times D_p} + 47.88 \times \left(\frac{D_{NO_3}}{\bar{c} \times D_p}\right)^2}$$

and $\gamma$ is the fraction of collisions between $NO_3$ and SOA leading to reaction, $D_p$ is the surface area-weighted mean particle diameter, $\rho_P$ is the particle density, $N_A$ is Avogadro's number, $c$ is the mean molecular speed of $NO_3$ ($3.2*10^4$ cm s$^{-1}$ at $T$ = 298 K), $M$ is the mean molecular weight of the SOA, and $D_{NO3}$ = 0.08 cm$^2$ s$^{-1}$ is the $NO_3$ diffusion coefficient in air (Rudich et al., 1996). Figure S4 shows $k_{SOA+NO3}$ as a function of $D_p$ ranging from 1 to 1000 nm assuming $\rho_P$ = 1.4 g cm$^{-3}$, $M_{SOA}$ = 250 g mol$^{-1}$ (Nah et al., 2016) and an upper limit $\gamma$ = 0.1 for BVOC-derived SOA (Ng et al., 2017). For reference, the range of slowest (isoprene) and fastest (humulene) $k_{BVOC+NO3}$ are indicated by the vertical blue line on the y-axis. In the limit where 100% of a BVOC is converted to SOA, $(NO_3R)_{ext}$ decreases by a factor of 10 or greater depending on $k_{BVOC+NO3}$ and $D_p$

Taken together, these results suggest that $(NO_3R)_{ext}$ decreases following $NO_3$ oxidation of BVOCs to carbonyl oxidation products and/or SOA. In this case, inputting $(NO_3R)_{ext}$ of the BVOC precursor to Eq. 5 generates a lower limit to $(NO_3)_{exp}$ over multiple generations of $NO_3$ oxidation. Results for other systems will depend on $k_{NO3}$ values of associated gas- and condensed-phase precursors and their oxidation products."

Table S5 and Figure S4 were added to the supplement:

**Table S5.** Bimolecular rate coefficients between selected biogenic volatile organic compounds (BVOCs) and $NO_3$, and BVOC + $NO_3$ carbonyl oxidation produts and $NO_3$. Rate coefficients were otained from Ng et al. (2017) and references therein, and are given in units of cm$^3$ molecules$^{-1}$ s$^{-1}$.

| BVOC | $k_{NO_3}$ | Oxidation Product | $k_{NO_3}$ |
|---|---|---|---|
| isoprene | $6.5\times10^{-13}$ | methyl vinyl ketone | $<6\times10^{-16}$ |
| | | methacrolein | $3.4\times10^{-15}$ |
| $\alpha$-pinene | $6.2\times10^{-12}$ | pinonaldehyde | $2.0\times10^{-14}$ |
| 3-carene | $9.1\times10^{-12}$ | caronaldehyde | $2.5\times10^{-14}$ |
| sabinene | $1.0\times10^{-11}$ | sabinaketone | $3.6\times10^{-16}$ |

[Figure]

**Figure S4.** Effective rate constant between $NO_3$ and SOA particles ($k_{NO_3}$) calculated using Eq. 6 assuming $\rho_P$ = 1.4 g cm$^{-3}$, $M_{SOA}$ = 250 g mol$^{-1}$ and $\gamma$ = 0.1.

The following citations were added to references:

N. A. Fuchs and A. G. Sutugin: Highly Dispersed Aerosols, Ann Arbor Science Publishers, Newton, MA, 1970.

A. T. Lambe, M. A. Miracolo, C. J. Hennigan, A. L. Robinson, and N. M. Donahue, Effective Rate Constants and Uptake Coefficients for the Reactions of Organic Molecular Markers (*n*-Alkanes, Hopanes, and Steranes) in Motor Oil and Diesel Primary Organic Aerosols with Hydroxyl Radicals, *Environ. Sci. Technol.* 43(23) 8794-8800, https://doi.org/10.1021/es901745h, 2009.

Y. Rudich, R. K. Talukdar, T. Imamura, R.W. Fox, and A.R. Ravishankara, Uptake of $NO_3$ on KI solutions: rate coefficient for the $NO_3$ + $I^-$ reaction and gas-phase diffusion coefficients for $NO_3$, *Chem. Phys. Lett.*, 261(4–5), 467-473, https://doi.org/10.1016/0009-2614(96)00980-3, 1996.

Specific comments:

2)      BBCES measuring the NO3:
   a.  The author stated that "I($\lambda$) and I0($\lambda$) were the measured transmitted intensities in the presence and absence of NO3". How was the "absence of NO3" achieved?

This is now specified in the text (p4, L16) (see below).

   b.  The equation grading the calculation of $\alpha(\lambda)$ is not correct. The $\alpha(\lambda)$ in the cavity also contributed by the bath gas beyond the NO3 radicals.

The equation is correct. Additional of information is now provided in the text to avoid any misunderstanding in the equation.

We modified the text as follows (changes bolded):

P4, L4: direct measurements of $NO_3$ generated via OR-i$N_2O_5$ were performed using a newly developed Incoherent Broad Band Cavity Enhanced Absorption Spectroscopy (IBBCEAS) technique (Cirtog et al., manuscript in preparation, **Fouqueau et al., 2020**).

P4, L11: Briefly, measurements were conducted by exciting a high-finesse optical cavity formed by two high reflectivity mirrors with an incoherent broad-band-source centered on the $\lambda$ = 662 nm absorption cross section of $NO_3$ **($2\times10^{-17}$ cm$^2$, Orphal et al., 2003).**

P4, L16: "**Where $\alpha(\lambda)$ is the absorption coefficient of the OFR sample in the instrument**, I($\lambda$) and I$_0$($\lambda$) were the measured transmitted intensities in the presence and absence **of the sample**, $d$ = 61 was the distance between the cavity mirrors, and $R(\lambda)$ was the mirror reflectivity (~99.98%). **I$_0$($\lambda$) was obtained by stopping the OFR sample through the instrument and flowing nitrogen from a cylinder (Air Liquide). A period of at least 30 s was allowed between the measurement of I$_0$($\lambda$) and I($\lambda$) to ensure the complete purge of the instrument. $R(\lambda)$** was measured before each experiment using a certified calibration cylinder containing 600 ppb $NO_2$ in zero air (Air Liquide)."

   c.  The NO3 radicals are highly reactive and can easily lose to the walls. What is the transmission efficiency of NO3 from the OFR to the cavity?
   d.   Due to different loss rates of $NO_3$ and $N_2O_5$ to the wall, the equilibrium of $NO_3$ and $N_2O_5$ may change. How good is the measured $NO_3$ concentration in the CRD represent the $NO_3$ radical concentration in the reactor?

We modified the text as follows (changes bolded):

P4, L20-27: " **Concentrations** were calculated **by applying a least square fit to the measured $\alpha(\lambda)$ considering the absorbing species in the sample:**

$$\alpha(\lambda) = [NO_2]\sigma_{[NO_2]}(\lambda) + [NO_3]\sigma_{[NO_3]}(\lambda) + [O_3]\sigma_{[O_3]} + p(\lambda)$$

**with NO$_2$, NO$_3$ and O$_3$ being the species absorbing in the spectral region of the instrument, σ(λ) are the respective absorption cross sections convoluted with the apparatus function (Vandaele et al., 1998; Orphal et al., 2003, Voigt et al, 2001) and p(λ) is a cubic polynomial to correct baseline deformations due to small LED intensity variations.** To avoid saturation of the IBBCEAS in these experiments, the OFR sample was diluted by a controlled dilution factor ranging from **9 to 41** and the detection response was deliberately lowered by reducing the optical path length. **Sampling lines and instrument (cavity) were made of PFA**. **The residence time in the IBBCEAS sampling line and instrument ranged from 8.3 to 21.8 s. At these residence times, the calculated transmission efficiency of NO$_3$ from the OFR to the IBBCEAS instrument ranged from 0.3 to 11% assuming a NO$_3$ wall loss rate constant of 0.27 s$^{-1}$ (Kennedy et al., 2011). Corrections to measured NO$_2$ and NO$_3$ values accounting for N$_2$O$_5$ thermal decomposition, N$_2$O$_5$ wall loss, and sample dilution in the IBBCEAS inlet were additionally applied to results presented in this paper.**"

The following citations were added to references:

Fouqueau, A., Cirtog, M., Cazaunau, M., Pangui, E., Zapf, P., Siour, G., Landsheere, X., Méjean, G., Romanini, D. and Picquet-Varrault, B.: Implementation of an IBBCEAS technique in an atmospheric simulation chamber for in situ NO3 monitoring: characterization and validation for kinetic studies, Atmos Meas Tech, (amt-2020-103), in review, 2020.

S. Voigt, J. Orphal, K. Bogumil, and J.P. Burrows, "The temperature dependence (203-293 K) of the absorption cross sections of O$_3$ in the 230-850 nm region measured by Fourier-transform spectroscopy", J. Photochem. Photobiol. A: Chem. 143, 1-9 (2001); DOI: 10.1016/S1010-6030(01)00480-4

3) a) The description of the results in section 3.4 is not consistent with the results in the figure. "First, at $[O_3]_{0,LFR}$< 1000 ppm and $[NO_2]_{0,LFR}$:$[O_3]_{0,LFR}$= 0.1 to 1.8, maximum $NO_{3exp}$ increased with decreasing $[NO_2]_{0,LFR}$:$[O_3]_{0,LFR}$ (Fig. 7a)." It is very hard to see the results in the figure when $[O_3]_{0,LFR}$< 100 ppm. For me, it looks like maximum $NO_{3exp}$ first increase with increasing $[NO_2]_{0,LFR}$:$[O_3]_{0,LFR}$ ratio and then decrease with it, especially when $[O_3]_{0,LFR}$ was in the range of 100-1000ppm"

We modified the text as follows:

"First, at $[O_3]_{0,LFR}$< 1000 ppm and $[NO_2]_{0,LFR}$:$[O_3]_{0,LFR}$= 0.**01** to 1.8, maximum $NO_{3exp}$ **increased with $[NO_2]_{0,LFR}$:$[O_3]_{0,LFR}$ prior to** decreasing **at $[NO_2]_{0,LFR}$:$[O_3]_{0,LFR}$ > 1.0** (Fig. 7a)."

b) "Above $[O_3]_{0,LFR}$≈ 2000 ppm, $NO_{3exp}$ was less sensitive to $[NO_2]_{0,LFR}$:$[O_3]_{0,LFR}$." This is true except for $[NO2]_{0,LFR}$:$[O_3]_{0,LFR}$=2.0.

We modified the text as follows:

"Above $[O_3]_{0,LFR}$≈ 2000 ppm **and below $[NO_2]_{0,LFR}$:$[O_3]_{0,LFR}$= 2.0**, $NO_{3exp}$ was less sensitive to $[NO_2]_{0,LFR}$:$[O_3]_{0,LFR}$."

c) "Second, maximum $NO_3$:$O_3$ increased with increasing $[NO_2]_{0,LFR}$:$[O_3]_{0,LFR}$ (Figure 7c)." This statement is true only when the $O_3$ was above 1000 ppm, even get rid of the results from $[NO_2]_{0,LFR}$:$[O_3]_{0,LFR}$=2.0.

We modified the text as follows:

"Second, maximum $NO_3$:$O_3$ increased with increasing $[NO_2]_{0,LFR}$:$[O_3]_{0,LFR}$ **above $[O_3]_{0,LFR}$ = 1000 ppm** (Figure 7c)."

d) "conversion of $O_3$ to $N_2O_5$ inside the LFR" I fell more comfortable to say "conversion of $O_3$ to $O_2$ inside the LFR".

We modified the text as follows:

"conversion of $O_3$ to **$O_2$** inside the LFR"

4) The authors tried to investigate the $RO_2$ fate and considered "$RO_2$ react with NO, $NO_2$, $NO_3$, $HO_2$, or other $RO_2$ to generate alkoxy (RO) radicals, peroxynitrates ($RO_2NO_2$), hydroperoxides or organic peroxides, and may additionally undergo autooxidation via sequential isomerization and $O_2$ addition." Recent studies by Berndt et al. (2018) revealed that self-and cross-reaction of $RO_2$ radicals would produce dimers effectively. How could this process affect the fate of the $RO_2$ radical?

Self- and cross-reactions of $RO_2$ were considered in the model – please see the last two rows in Table S3, reproduced below for reference. Under the conditions that were studied this process was minor compared to $RO_2$ + $NO_3$ and $RO_2$ + $NO_2$.

**Table S3.** KinSim mechanism used to model destruction of alkyl and acyl organic peroxy radicals formed from VOC + $NO_3$ reactions in the OFR. Kinetic data is adapted from (Orlando and Tyndall, 2012).

| Reactant 1 | Reactant 2 | Product 1 | Product 2 | Product 3 | $A_\infty$ | $E_\infty$ | $n_\infty$ | $A_0$ | $E_0$ | $n_0$ |
|---|---|---|---|---|---|---|---|---|---|---|
| VOC | $NO_3$ | alkylRO2 | | | 0 or 2.5E-12 | 0 | 0 | 0 | 0 | 0 |
| VOC | $NO_3$ | acylRO2 | | | 2.5E-12 or 0 | 0 | 0 | 0 | 0 | 0 |
| alkylRO2 | NO | | | | 2.7E-12 | -360 | 0 | 0 | 0 | 0 |
| acylRO2 | NO | | | | 7.5E-12 | -290 | 0 | 0 | 0 | 0 |
| alkylRO2 | NO2 | alkylRO2NO2 | | | 6.1E-12 | 0 | 0 | 1.3E-30 | 6.2 | 0.31 |
| alkylRO2NO2 | | alkylRO2 | NO2 | | 8.8E+15 | 10440 | 0 | 0.00048 | 9285 | 0.31 |
| acylRO2 | NO2 | acylRO2NO2 | | | 1.2E-11 | 0 | 0.9 | 2.7E-28 | 7.1 | 0.3 |
| acylRO2NO2 | | acylRO2 | NO2 | | 5.4E+16 | 13830 | 0 | 0.0049 | 12100 | 0.3 |
| alkylRO2 | NO3 | alkylRO | NO2 | | 2.4E-12 | 0 | 0 | 0 | 0 | 0 |
| acylRO2 | NO3 | acylRO | NO2 | | 3.2E-12 | 0 | 0 | 0 | 0 | 0 |
| alkylRO2 | HO2 | | | | 7.4E-13 | -700 | 0 | 0 | 0 | 0 |
| acylRO2 | HO2 | | | | 5.2E-13 | -980 | 0 | 0 | 0 | 0 |
| alkylRO2 | acylRO2 | | | | 2.2E-12 | -500 | 0 | 0 | 0 | 0 |
| acylRO2 | acylRO2 | | | | 2.9E-12 | -500 | 0 | 0 | 0 | 0 |

In response to the reviewer's comment, we conducted additional sensitivity studies where the $RO_2$ + $RO_2$ rate constant was assumed to be $1 \times 10^{-10}$ $cm^3$ $molecule^{-1}$ $s^{-1}$, close to the values reported by Berndt et al. (2018). The sensitivity cases cover the range of conditions used in Section 3.4 and those with $NO_3R_{ext}$ (and

hence VOC concentration) increased by a factor of 10. In these cases, the relative contribution of $RO_2$ + $RO_2$ to $RO_2$ fate is always <1%. To include this information in the paper, we modified the text as follows:

P8, L23-25: "Under almost all OFR-i$N_2O_5$ conditions shown in Figure 7, $RO_2$ reactions with NO, $HO_2$, and $RO_2$ were minor (< 1%) loss pathways compared to reaction with $NO_2$ and $NO_3$. **We conductive a model sensitivity analysis in which the $RO_2$ + $RO_2$ reaction rate was enhanced by increasing $NO_3R_{ext}$ from 0.07 to 0.7 s$^{-1}$ and increasing the $RO_2$ + $RO_2$ rate constant from 1x10$^{-11}$ to 1x10$^{-10}$ cm$^3$ molecule$^{-1}$ s$^{-1}$ (Berndt et al., 2018, 2018a). Despite these perturbations, the relative contribution of $RO_2$ + $RO_2$ reactions to total $RO_2$ loss remained < 1% across this range of OFR-i$N_2O_5$ conditions.**"

The following citations were added to References:

Berndt, T.; Mentler, B.; Scholz, W.; Fischer, L.; Herrmann, H.; Kulmala, M.; Hansel, A., Accretion Product Formation from Ozonolysis and OH Radical Reaction of alpha-Pinene: Mechanistic Insight and the Influence of Isoprene and Ethylene. Environ. Sci. Technol. 2018, 52, (19), 11069-11077.

Berndt, T.; Scholz, W.; Mentler, B.; Fischer, L.; Herrmann, H.; Kulmala, M.; Hansel, A., Accretion Product Formation from Self- and Cross-Reactions of RO2 Radicals in the Atmosphere. Angew Chem Int Edit 2018, 57, 1, 3820-3824.

**Anonymous Referee #2 – N/A (referee did not submit review)**

**Anonymous Referee #3**

General comments.

1.      a) Section 2.2.1, the section cited some references about the wall loss of NO3 and N2O5 in Teflon/Pyrex tube, and the NO3 and N2O5 wall loss in LFR and OFR is extrapolating or interpolation based on the reported results, which weak the results as the wall loss of NO3 and N2O5 in the system is an important source of uncertainty. The lab quantification of the wall loss in LFR and OFR in the future works can further improve the value of this study.

Our results suggest that $NO_3$ wall loss is not an important source of uncertainty because $NO_3$ is too short-lived for $NO_3$ wall loss to compete with $NO_3$ oxidative loss. We modified the text as follows to underscore the importance of characterizing $N_2O_5$ wall loss rates for a specific LFR-OFR configuration (changes bolded):

P12, L8-12: "Because OFR-i$N_2O_5$ can continuously generate $N_2O_5$ and $NO_3$ at room temperature, it is significantly easier to apply in continuous flow reactor studies than related techniques. However, in addition to the aforementioned considerations, high $N_2O_5$ and $HNO_3$ concentrations that are generated using OFR-i$N_2O_5$ complicate the application of techniques such as iodide-adduct chemical ionization mass spectrometry due to efficient reactions between the iodide reagent ion and $N_2O_5$ or $HNO_3$ (Lee et al.,2014). **Additionally, the humidity-dependent $N_2O_5$ wall loss rate must be accurately characterized to model the performance of a specific OFR-i$N_2O_5$ configuration.**"

b) Additionally, page 5, line 9-10, the fixed condition of OFR is ambiguous, please clear it, at least add the simulation results in SI.

We modified the text as follows (changes bolded):

P5, L6-10: "Published $k_{wLFR,NO3}$ values onto tubing with 1 cm **(Teflon)** and 4 cm **(Pyrex)** ID are 0.2 and 0.1 s$^{-1}$ respectively [...] Assuming $k_w$ is inversely proportional to the internal diameter of the tube, we assumed $k_{wLFR,NO3}$= 0.15 s$^{-1}$. Extrapolating this value to the OFR yielded $k_{wOFR,NO3}$ = 0.02 s$^{-1}$. At fixed OFR-iN$_2$O$_5$ conditions **that are summarized in Table S3**, varying $k_{wLFR,NO3}$ between 0 and 0.**3** s$^{-1}$ changed NO$_{3exp}$ achieved in the OFR by 0.3%."

We added the following table to the Supplement:

**Table S3.** Sensitivity analysis of the effect of varying $k_{w_{LFR},NO_3}$ on NO$_{3exp}$. The following inputs to the KinSim mechanism were assumed: $[NO_2]_{0,LFR} = [O_3]_{0,LFR} = 300$ ppm, $T_{LFR} = T_{OFR} = 24°C$, $RH_{LFR} = RH_{OFR} = 1\%$, $k_{w_{LFR},N_2O_5} = 0.1$ s$^{-1}$, $k_{w_{OFR},N_2O_5} = 0.014$ s$^{-1}$, $\tau_{LFR} = 20$ s, $\tau_{OFR} = 120$ s, dilution factor = 4.4 between LFR and OFR.

| $k_{w_{LFR},NO_3}$ [s$^{-1}$] | NO$_{3exp}$ [molecules cm$^{-3}$ s] | Normalized NO$_{3exp}$ |
|---|---|---|
| 0 | $1.277 \times 10^{14}$ | 1 |
| 0.1 | $1.275 \times 10^{14}$ | 0.9984 |
| 0.2 | $1.273 \times 10^{14}$ | 0.9969 |
| 0.3 | $1.272 \times 10^{14}$ | 0.9965 |

c) Page 5, line 11-13, the Extrapolating results is confuse, the reference said 0.04 and 0.009 s-1 corresponding to ID (4 and 7 cm), what is corresponding parameter of the kwall,LFR of 0.07 and 0.03 s-1 mentioned here?

We assumed $k_{wLFR,N2O5}$ = 0.05 s$^{-1}$ by calculating the average of the cited $k_{wLFR,N2O5}$ = 0.03 and 0.07 s$^{-1}$ values that were obtained after extrapolating to the ID of the LFR. We clarified this by modifying the text as follows (changes bolded):

P5, L11-13: "Published k$_{w,N2O5}$ values onto dry (RH≈20%) Pyrex/PFA tubing with 4 and 7 cm ID are 0.04 and 0.009 s$^{-1}$ [...] Extrapolating these values to the LFR used here **and then averaging them together yielded** k$_{w,N2O5}$= 0.05 s$^{-1}$ **that was applied** in the LFR-KinSim model."

Specific comments.

2.  Page 2, line 29-34, this introduction of the LFR is confusing. The authors can use a schematic figure to show more details about the OFR-iN2O5 (rather than Figure 1 from references), which would increase the paper's readability.

Figure 1 in this manuscript, reproduced below, already shows the OFR-iN$_2$O$_5$ schematic that we think the reviewer is requesting:

[Figure]

**Figure 1.** Process flow diagram of the OFR-iN$_2$O$_5$ technique used to generate nitrate radicals (NO$_3$).

We are assuming that the references to Wood et al. (2003) and Boyd et al. (2015) caused the confusion, based on the reviewer's statement regarding "Figure 1 from references". To clarify this section, we modified the text as follows (changed bolded):

P2, L29-31: "Figure 1 shows a process flow diagram of the OFR-iN$_2$O$_5$ method. Separate flows containing NO$_2$ and O$_3$ were input to a PFA tube with 2.54 cm outer diameter, 2.22 cm inner diameter, and 152.4 cm length that was operated as an LFR. **Previous studies used a similar process to generate N$_2$O$_5$** (Wood et al., 2003; Boyd et al., 2015) **although the LFR materials, flow rates, and reagent concentrations were different.**"

    3.  Page 4, line 8. Romanini et al. (1997) is not the right reference of the IBBCEAS principle, I suggest the author replace it by e.g., Fiedler et al., 2003.(Fiedler, S. E., Hese, A., and Ruth, A. A.: Incoherent broad-band cavity-enhanced absorption spectroscopy, Chem Phys Lett, 371, 284-294, 2003.)

We made the substitution suggested by the reviewer.

    4.  In Eq .2 the kNO2 or kNO3 should revised to kNO2+RO2 or kNO3+RO2, the similar change also applied in Eq. 3 and Eq. 4

We made the revisions suggested by the reviewer.

[revised manuscript text omitted]
$_{3\text{exp}}$ model results was defined as follows: [O$_3$]$_{0,\text{LFR}}$ = 10-1000 ppm, [NO$_2$]$_{0,\text{LFR}}$ = 10-1000 ppm, [NO$_2$]$_{0,\text{LFR}}$:[O$_3$]$_{0,\text{LFR}}$ $\leq$ 2, NO$_3$R$_{\text{ext}}$ = 1-200 s$^{-1}$, $k_{\text{w}_{\text{OFR}},\text{N}_2\text{O}_5}$ = 0.01-0.08 s$^{-1}$, $T_{\text{OFR}}$ = 0 - 40°C, and $\tau_{\text{OFR}}$ = 60 - 300 s. The cases where [O$_3$]$_{0,\text{LFR}}$ > 1000 ppm and/or [NO$_2$]$_{0,\text{LFR}}$:[O$_3$]$_{0,\text{LFR}}$ > 2 were not considered because of less practical interest. We explored 11, 11, 7, 4, and 5 logarithmically evenly distributed values in the ranges of [O$_3$]$_{0,\text{LFR}}$, [NO$_2$]$_{0,\text{LFR}}$ (11 values over 10–1000 ppm), NO$_3$R$_{\text{ext}}$, $k_{\text{w},\text{N}_2\text{O}_5}$, and $\tau_{\text{OFR}}$, respectively. Due to significantly different chemical regimes in different parts of the phase space, fit coefficients that are reported in Table 1 were obtained by fitting the same functional form (Equation 6) over 3 sub-phase spaces with the following additional constraints: (i1) [NO$_2$]$_{0,\text{LFR}}$:[O$_3$]$_{0,\text{LFR}}$ = 0-1 and NO$_3$R$_{\text{ext}}$ = 20-200 s$^{-1}$; (ii2) [NO$_2$]$_{0,\text{LFR}}$:[O$_3$]$_{0,\text{LFR}}$ = 0-1 and NO$_3$R$_{\text{ext}}$ = 1-20 s$^{-1}$ (iii3) [NO$_2$]$_{0,\text{LFR}}$:[O$_3$]$_{0,\text{LFR}}$ = 1-2. For these 3 subspaces, 10080, 13440, and 5880 model cases respectively were simulated. In Equation 6, the terms involving the coefficients $g$–$j$ were included to reproduce the relationship between normalized NO$_{3\text{exp}}$ and [NO$_2$]$_{0,\text{LFR}}$:[O$_3$]$_{0,\text{LFR}}$ shown in Figure 5. Logarithms of first- and second-order terms were successively added until no further fit quality improvement was achieved. Figure 10 compares NO$_{3\text{exp}}$ estimated from Equation 6 and calculated from the model described in Section 2.2. The mean absolute value of the relative deviation was 49% which is comparable with results obtained for previous estimation equations with significant NO$_\text{y}$ chemistry (Peng et al., 2018).

NO$_3$R$_{\text{ext}}$ of a system will change over the course of multiple generations of NO$_3$ oxidation due to changes in kinetic rate coefficients between different species and NO$_3$ ($k_{\text{NO}_3}$). The sensitivity of Eq. 6 to changes in NO$_3$R$_{\text{ext}}$ depends in part on the relative magnitudes of NO$_3$R$_{\text{ext}}$ and the internal NO$_3$ reactivity, NO$_3$R$_{\text{int}}$, which is approximately equal to $k_{\text{NO}_2+\text{NO}_3}$[NO$_2$]. If NO$_3$R$_{\text{int}}$ >> NO$_3$R$_{\text{ext}}$, changes in NO$_3$R$_{\text{ext}}$ would have minimal influence on Eq. 6.

In one case study, we examined changes in NO$_3$R$_{\text{ext}}$ following conversion of biogenic VOCs (BVOCs) to gas-phase carbonyl oxidation products with known $k_{\text{NO}_3}$ values. Table S5 compares $k_{\text{NO}_3}$ of isoprene to methyl vinyl ketone and methacrolein, $\alpha$-pinene to pinonaldehyde, sabinene to sabinaketone, and 3-carene to caronaldehyde. In the limit where 100% of each BVOC is converted to its carbonyl oxidation product, NO$_3$R$_{\text{ext}}$ decreases by a factor of 200 or greater. Unsaturated organic nitrates that are generated from BVOC + NO$_3$ may also be reactive towards NO$_3$, but $k_{\text{NO}_3}$ for these species are not available. In another case study, we examined changes in NO$_3$R$_{\text{ext}}$ following conversion of BVOCs to SOA. An effective $k_{\text{NO}_3}$ for SOA was calculated using the following equation adapted from Lambe et al. (2009):

$$k_{\text{NO}_3} = \frac{3}{2} \frac{\gamma \times \bar{c} \times M_{\text{SOA}} \times F_{\text{diff}}}{D_\text{p} \times \rho_\text{p} \times N_\text{A}} \tag{7}$$

where $F_{\text{diff}}$ is a correction factor accounting for diffusion limitations to the particle surface in the transition regime (Fuchs and Sutugin, 1970):

$$F_{\text{diff}} = \frac{1 + 6 \times \frac{D_{\text{NO}_3}}{\bar{c} \times D_\text{p}}}{1 + 10.26 \times \frac{D_{\text{NO}_3}}{\bar{c} \times D_\text{p}} + 47.88 \times \left(\frac{D_{\text{NO}_3}}{\bar{c} \times D_\text{p}}\right)^2} \tag{8}$$

and $\gamma$ is the fraction of collisions between NO$_3$ and SOA resulting in reaction, $D_\text{p}$ is the surface area-weighted mean particle diameter, $\rho_\text{p}$ is the particle density, $N_\text{A}$ is Avogadro's number, $\bar{c}$ is the mean molecular speed of NO$_3$ (3.2×10$^4$ cm s$^{-1}$ at

[revised manuscript text omitted]

than $OH_{exp}$, and (3)  $(NO_3R)_{int}$ of OFR-i$N_2O_5$, which is dominated by the $NO_3 + NO_2$ reaction, is larger and easier to manipulate than the internal OH reactivity of OFR-i$N_2O$ and OFR-i$C_3H_7ONO$, which is dominated by $OH + HO_2$ and $OH + NO_2$ reactions. To identify optimal OFR-i$N_2O_5$ conditions for different applications, we characterized
5  $NO_{3exp}$, $\tau_{RO_2}$, $F_{RO_2+NO_3}$, $F_{R+O_2}$ and $F_{VOC+NO_3}$ at $[O_3]_{0,LFR}$ = 10 ppm to 10%, $[NO_2]_{0,LFR}$:$[O_3]_{0,LFR}$ = 0.01 to 2.0, and $RH_{OFR}$ = 7 to 85%. Optimal $NO_{3exp}$ was achieved by minimizing $[H_2O]$ in the OFR and associated humidity-dependent $N_2O_5$ wall losses. This is contrary to most OFR techniques that are used to generate OH radicals, where optimal $OH_{exp}$ is achieved by maximizing $[H_2O]$ and associated OH production from the $O(^1D) + H_2O$ reaction and/or $H_2O$ 
[revised manuscript text omitted]

[Figure]

**Figure S1.** Residence time distribution of 10 s pulsed inputs of $NO_2$ injected into the Potential Aerosol Mass OFR obtained with lights off and 6.5 L min$^{-1}$ flow through the reactor.

[Figure]

**Figure S2.** Model simulations of the relative $NO_{3exp}$ achieved in the OFR following injection of 300 ppm $O_3$ and $NO_2$ into the LFR as a function of $\tau_{LFR}$ ranging from 1 to 60 s. Purple and red lines represent modeling cases corresponding to 0% ("no heating") and 100% ("heating") thermal dissociation of $N_2O_5$ between the LFR and OFR.

[Figure]

**Figure S3.** Relative rate constant obtained from PTR-MS measurements of butanal ($C_4H_8O$) and thiophene ($C_4H_4S$) tracers used in OFR-i$N_2O_5$ characterization studies. Literature relative rate constant obtained from kinetic data published by Atkinson (1991) and D'Anna et al. (2001).

[Figure]

**Figure S4.**  Effective rate constant between $NO_3$   and SOA particles (b$k_{NO_3}$) calculated using Eq.   6 assuming $\rho_P = 1.4$ g cm$^{-3}$, $M_{SOA} = 250$ g mol$^{-1}$ and $\gamma = 0.1$.

[Figure]

**Figure S5.** Logarithmically scaled AMS spectra of SOA generated from $NO_3$ oxidation of $\beta$-pinene in the (a) Georgia Tech environmental chamber (Boyd et al., 2015) and (b) OFR. Scatter plot in (c) shows spectra generated in the OFR and chamber plotted against each other.

**Table S1.** VOC tracers used in OFR-iN$_2$O$_5$ characterization studies. Bimolecular rate  coefficients for reaction with NO$_3$ and O$_3$ are given in units of cm$^3$ molecule$^{-1}$ s$^{-1}$.

| Compound | Formula | Structure | $k_{NO_3}$ | $k_{O_3}$ | References |
|---|---|---|---|---|---|
| Acetonitrile | C$_2$H$_3$N |  | $<3.01\times10^{-19}$ | N/A | 1 |
| Toluene | C$_7$H$_8$ |  | $6.79\times10^{-17}$ | $3.90\times10^{-22}$ | 2,3 |
| o-Xylene | C$_8$H$_{10}$ |  | $3.77\times10^{-16}$ | $1.72\times10^{-21}$ | 2,3 |
| p-Cymene | C$_{10}$H$_{14}$ |  | $1.00\times10^{-15}$ | $<5.00\times10^{-20}$ | 2,4 |
| 1,2,4-Trimethylbenzene | C$_9$H$_{12}$ |  | $1.81\times10^{-15}$ | $<1.3\times10^{-21}$ | 2 |
| Butanol | C$_4$H$_{10}$O |  | $<2.71\times10^{-15}$ | N/A | 5 |
| Benzaldehyde | C$_7$H$_6$O |  | $4.3\times10^{-15}$ | $<2.00\times10^{-19}$ | 2,6 |
| Butanal | C$_4$H$_8$O |  | $1.22\times10^{-14}$ | N/A | 7 |
| Thiophene | C$_4$H$_4$S |  | $3.94\times10^{-14}$ | $5.99\times10^{-20}$ | 2,8 |
| 2,3-Dihydrobenzofuran | C$_8$H$_8$O |  | $1.15\times10^{-13}$ | $<1.00\times10^{-19}$ | 2,9 |
| Naphthalene-d8 | C$_{10}$D$_8$ |  | $4.76\times10^{-28}\times[NO_2]$ | N/A | 2 |

[1]Cantrell et al. (1987); [2]Atkinson (1991); [3]Toby et al. (1985); [4]Atkinson et al. (1990); [5]Chew et al. (1998); [6]Bernard et al. (2013); [7]D'Anna et al. (2001); [8]Atkinson et al. (1983); [9]Atkinson et al. (1992)

**Table S2.** KinSim mechanism used to model $NO_3$ and $N_2O_5$ formation and destruction in the LFR and OFR. Kinetic data is adapted from mechanism published in Palm et al. (2017) and references therein.

| Reactant 1 | Reactant 2 | Product 1 | Product 2 | Product 3 | $A_\infty$ | $E_\infty$ | $n_\infty$ | $A_0$ | $E_0$ | $n_0$ |
|---|---|---|---|---|---|---|---|---|---|---|
| NO | $O_3$ | $NO_2$ | $O_2$ | | 3E-12 | 1500 | 0 | 0 | 0 | 0 |
| $NO_2$ | $O_3$ | $NO_3$ | $O_2$ | | 1.2E-13 | 2450 | 0 | 0 | 0 | 0 |
| $NO_3$ | $NO_3$ | $NO_2$ | $NO_2$ | $O_2$ | 8.5E-13 | 2450 | 0 | 0 | 0 | 0 |
| $N_2O_5$ | $H_2O$ | $HNO_3$ | $HNO_3$ | | 1E-22 | 0 | 0 | 0 | 0 | 0 |
| $NO_2$ | $NO_3$ | $N_2O_5$ | | | 1.9E-12 | 0 | -0.2 | 3.6E-30 | 0 | 4.1 |
| $N_2O_5$ | | $NO_2$ | $NO_3$ | | 9.7E+14 | 11080 | -0.1 | 0.0013 | 11000 | 3.5 |
| NO | $NO_3$ | $NO_2$ | $NO_2$ | | 1.8E-11 | -110 | 0 | 0 | 0 | 0 |
| $NO_2$ | $NO_3$ | NO | $NO_2$ | $O_2$ | 4.5E-14 | 1260 | 0 | 0 | 0 | 0 |
| $NO_3$ | wall1 | wall1-$NO_3$ | | | 0.02 - 0.15 | 0 | 0 | 0 | 0 | 0 |
| $N_2O_5$ | wall2 | wall2-$N_2O_5$ | | | 0.01 - 0.08 | 0 | 0 | 0 | 0 | 0 |

**Table S3.** Sensitivity analysis of the effect of varying $k_{w,LFR,NO_3}$ on $NO_{3exp}$. The following inputs to the KinSim mechanism were assumed: $[NO_2]_{0,LFR} = [O_3]_{0,LFR} = 300$ ppm, $T_{LFR} = T_{OFR} = 24°C$, $RH_{LFR} = RH_{OFR} = 1\%$, $k_{w,LFR,N_2O_5} = 0.1$ s$^{-1}$, $k_{w,OFR,N_2O_5} = 0.014$ s$^{-1}$, $\tau_{LFR} = 20$ s, $\tau_{OFR} = 120$ s, dilution factor = 4.4 between LFR and OFR.

| $k_{w,LFR,NO_3}$ [s$^{-1}$] | $NO_{3exp}$ [molecules cm$^{-3}$ s] | Normalized $NO_{3exp}$ |
|---|---|---|
| 0 | $1.277\times10^{14}$ | 1 |
| 0.1 | $1.275\times10^{14}$ | 0.9984 |
| 0.2 | $1.273\times10^{14}$ | 0.9969 |
| 0.3 | $1.272\times10^{14}$ | 0.9965 |

**Table S4.** KinSim mechanism used to model destruction of alkyl and acyl organic peroxy radicals formed from VOC + NO$_3$ reactions in the OFR. Kinetic data is adapted from Orlando and Tyndall (2012).

| Reactant 1 | Reactant 2 | Product 1 | Product 2 | Product 3 | A$_\infty$ | E$_\infty$ | n$_\infty$ | A$_0$ | E$_0$ | n$_0$ |
|---|---|---|---|---|---|---|---|---|---|---|
| VOC | NO$_3$ | alkylRO2 | | | 0 or 2.5E-12 | 0 | 0 | 0 | 0 | 0 |
| VOC | NO$_3$ | acylRO2 | | | 2.5E-12 or 0 | 0 | 0 | 0 | 0 | 0 |
| alkylRO2 | NO | | | | 2.7E-12 | -360 | 0 | 0 | 0 | 0 |
| acylRO2 | NO | | | | 7.5E-12 | -290 | 0 | 0 | 0 | 0 |
| alkylRO2 | NO2 | alkylRO2NO2 | | | 6.1E-12 | 0 | 0 | 1.3E-30 | 6.2 | 0.31 |
| alkylRO2NO2 | | alkylRO2 | NO2 | | 8.8E+15 | 10440 | 0 | 0.00048 | 9285 | 0.31 |
| acylRO2 | NO2 | acylRO2NO2 | | | 1.2E-11 | 0 | 0.9 | 2.7E-28 | 7.1 | 0.3 |
| acylRO2NO2 | | acylRO2 | NO2 | | 5.4E+16 | 13830 | 0 | 0.0049 | 12100 | 0.3 |
| alkylRO2 | NO3 | alkylRO | NO2 | | 2.4E-12 | 0 | 0 | 0 | 0 | 0 |
| acylRO2 | NO3 | acylRO | NO2 | | 3.2E-12 | 0 | 0 | 0 | 0 | 0 |
| alkylRO2 | HO2 | | | | 7.4E-13 | -700 | 0 | 0 | 0 | 0 |
| acylRO2 | HO2 | | | | 5.2E-13 | -980 | 0 | 0 | 0 | 0 |
| alkylRO2 | acylRO2 | | | | 2.2E-12 | -500 | 0 | 0 | 0 | 0 |
| acylRO2 | acylRO2 | | | | 2.9E-12 | -500 | 0 | 0 | 0 | 0 |

**Table S5.** Bimolecular rate coefficients between selected biogenic volatile organic compounds (BVOCs) and NO$_3$, and BVOC + NO$_3$ carbonyl oxidation produts and NO$_3$. Rate coefficients were otained from Ng et al. (2017) and references therein, and are given in units of cm$^3$ molecule$^{-1}$ s$^{-1}$.

| BVOC | k$_{NO_3}$ | Oxidation Product | k$_{NO_3}$ |
|---|---|---|---|
| isoprene | $6.5\times10^{-13}$ | methyl vinyl ketone | $<6\times10^{-16}$ |
| | | methacrolein | $3.4\times10^{-15}$ |
| $\alpha$-pinene | $6.2\times10^{-12}$ | pinonaldehyde | $2.0\times10^{-14}$ |
| 3-carene | $9.1\times10^{-12}$ | caronaldehyde | $2.5\times10^{-14}$ |
| sabinene | $1.0\times10^{-11}$ | sabinaketone | $3.6\times10^{-16}$ |